# Dynamic maps of people exposure to floods based on mobile phone data

Matteo Balistrocchi<sup>1</sup>, Rodolfo Metulini<sup>2</sup>, Maurizio Carpita<sup>3</sup>, Roberto Ranzi<sup>4</sup>

<sup>1</sup>Department of Engineering "Enzo Ferrari", University of Modena and Reggio Emilia, Modena (MO), 41125, Italy
 <sup>2</sup>Department of Economics and Statistics, University of Salerno, Fisciano (SA), 84084, Italy
 <sup>3</sup>Department of Economics and Management, University of Brescia, Brescia (BS), 25122, Italy

<sup>4</sup>Department of Civil, Environmental, Architectural Engineering and Mathematics, University of Brescia, Brescia (BS), 25123, Italy

Correspondence to: Matteo Balistrocchi (matteo.balistrocchi@unimore.it)

**Abstract.** Floods are acknowledged as one of the most serious threats to people's lives and properties worldwide. To mitigate the flood risk, it is possible to act separately on its components: hazard, vulnerability, exposure. Emergency management plans can actually provide effective non-structural practices to decrease both people exposure and vulnerability. Crowding maps depending on characteristic time patterns, herein referred to as dynamic exposure maps, represent a valuable tool to enhance

- the flood risk management plans. In this paper, the suitability of mobile phone data to derive crowding maps is discussed. A test case is provided by a strongly urbanized area subject to frequent floodings located on the western outskirts of Brescia (northern Italy). Characteristic exposure spatio-temporal patterns and their uncertainties were detected, with regard to land cover and calendar period. This novel methodology still deserves verification during real-world flood episodes, even though it appears to be more reliable than crowdsourcing strategies, and seems to have potentials to better address real-time rescues
- and relief supplies.

## **1** Introduction

Floods are natural phenomena whose hazards afflict nearly 20 million people worldwide (Kellens et al., 2013), posing a serious challenge to the protection of people's lives and the liveability of urban settlements. A high-confidence increase trend in the economic damages and social costs due to extreme weather events has globally been documented (Kreibich et al., 2019). Two

- major factors can be advocated for justifying such a trend: climate change and increased urbanization (Hartmann et al., 2013). These factors involve different components of the risk concept (UN ISDR, 2009), which is given by the combination of hazard, exposure and vulnerability. Climate change was popularly acknowledged as a leading cause for the increases in the frequency and intensity of heavy storms and, consequently, of the flood hazard (Solomon et al., 2007). However, according to the Intergovernmental Panel on Climate Change (IPCC) (Hartmann et al., 2013), and as confirmed also by up-to-date analyses of the increase of
- flood intensity in Europe (Blöschl et al., 2019), the absence of a global likely trend in the incidence of floods arises. Reasons

lie in the high regional variability of heavy storm trends, as well as in the strong influence played by watershed hydrologic characteristics and local flood management practices.

On the contrary, population urbanization represents a likely global trend, though characterized by a strong regional variability. Migration from countryside or mountain areas to cities is the main driver of urban sprawl. In 2008, for the first time in human

- history, more than half the world population was living in urban settlements and the percentage continues to augment (UN DESA Population Division, 2012). Touristic demand is an additional driver for urbanization growth, that plays a peculiar role in developed countries. For instance, in Italy many areas are affected by emigration, namely Alpine and Apennine valleys, southern regions and islands, and urbanization growth rates are equal to or greater than the national average rate. A clear example is provided by Southern Italy, where the annual rate of soil consumption between 2017 and 2018 was 0.23 %, greater
- than the national average of 0.21 % (ISPRA, 2019), even if Southern Italy faced a population decrease of 1.5 % between 2015 and 2019 (the national average loss was estimated at 0.7 %) (CENSIS, 2019). An additional example is provided by the Sondrio province (in the mountainous part of the Adda River basin, Lombardy, Northern Italy), where the yearly soil consumption per capita is 1.11 m<sup>2</sup>, whereas the regional one is 0.63 m<sup>2</sup>.

Urbanization determines dramatic increases in people exposure and vulnerability to floods in immigration areas, since most of

- recent urbanizations lie in flood prone areas and local communities are not able to put effective flood defence practices in place. Moreover, urbanization usually leads to the impairment of the conveyance capacity of the stream network, so that flooding areas are basically larger than in the undeveloped condition. The urbanization sprawl consequently results in increased damage to communities, private properties and public infrastructures, and must be regarded as the main cause for the likely increasing trend of flood risk (Barredo et al., 2009). A number of researches on flood risk changes under economic and
- population growth scenarios indicate that this contribution is at least equal to, but commonly larger than the climate change one (Feyen et al., 2009; Maaskant et al., 2009; Bouwer et al., 2010; Te Linde et al., 2011; Rojas et al., 2013). This flood risk trend forced an unavoidable shift in the paradigms of flood defence, by recognizing that not all events can be completely controlled and that structural practices have limits (Johnson and Priest, 2008). Thus, focus must be placed on how to mitigate the damages to flood prone communities (European Union, 2007). Over the last decades, flood risk management
- has evolved from a structural-based defence approach, aiming at decreasing the hazard component, towards a more holistic perspective (Merz et al., 2010; Arrighi et al., 2019) taking into consideration vulnerability and exposure. Novel concepts were introduced, such as residual risk (UN ISDR, 2009), accounting for the potential structural failure of the defence system (Vorogushin et al., 2009; Schumann, 2017; Balistrocchi et al., 2019), and resilience, that is the ability to recover from a damage or to absorb an impact (Liao, 2012).
- Differently from the hazard, the mitigation of exposure and vulnerability can be pursued by means of non-structural practices. Among them, a prime role is played by emergency management plans, which allow authorities responsible for the protection of local communities to dispatch timely and appropriate mitigation measures during the occurrence of flood episodes. The development of effective emergency management plans is closely related to the concept of authorities' preparedness (European Union, 2007). Such plans are actually intended to provide people with early warnings, reliable real-time information and the

improvement of relief supplies and rescue efforts. In this regard, a detailed and reliable picture of the real-time spatiotemporal variability of the flood risk would be highly beneficial.

Presently, large amounts of geospatial data can be obtained from a number of sources, namely remote sensing or aircraft platforms. These sources yield situation snapshots, but do not provide information at the spatiotemporal resolution needed for managing urban floodings and are hardly validated on the field. To overcome these problems, the possibility of taking

- advantage of crowdsourcing techniques has recently attracted much attention (Mazumdar et al., 2017; Rosser et al., 2017; Hirata et al., 2018; Mazzoleni et al., 2018). These techniques have been made available by the widespread proliferation of smartphones and tablets, along with the success of social media. During emergency phases, the advantages in the real-time implementation of the emergency plan are twofold: firstly, a large amount of volunteered geographical information suitably georeferenced can be collected (Goodchild, 2007), guiding authorities in developing collaborative flooding maps or in
- estimating the number and location of exposed people; secondly, crucial information can be communicated to the people exposed, making them aware of the actual risk magnitude and enhancing their capability to face the situation. After the emergency phase, authorities can also exploit the collected data to enhance their preparedness and to better match the emergency management plans to specific real-world needs of the flood prone community.
- Potentials and drawbacks of various crowdsourcing techniques have long been debated, even though crowdsourcing has already found successful applications in some weather related disasters (Poser and Dransch, 2010; Hung et al., 2016; Guntha et al., 2018). Such researches have underlined that in many cases, during the emergency phase, crowdsourced data have at least the same quality as the authoritative ones (Goodchild et al., 2017). Nevertheless, several concerns have been pointed out through analyses of crowdsourcing technique applications to real-world disasters: i) the raw data quality is generally poor because of malicious intentions of nasty elements in the community or incompetence of stakeholders, so that spurious,
- erroneous, malformed, redundant or incomplete data must be purged out of the database to make them suitable, ii) the sample significance is basically low, owing to the limited number of exposed people actively participating in crowdsourcing, so that information of general interest must be extrapolated from an exiguous fraction of the whole population, iii) the communication network is not completely reliable, as it could fail or malfunction during disaster occurrences.
- Researches have therefore been addressed to filter such rumours from crowdsourced data (Han and Ciravegna, 2019). 90 However, other approaches can be followed to develop effective dynamic information tools through the exploitation of mobile phone data collected by providers. Such data make it possible to geo-localize mobile phone users over time, in order to derive time-dependent crowding maps. When such maps are intersected with hazard maps, showing the flooding area extensions corresponding to a selected frequency of occurrence, dynamic exposure maps are obtained. As demonstrated by Carpita and Simonetto (2014) with reference to episodic crowd concentrations due to social events, recurrent spatiotemporal patterns can
- be derived from mobile phone data by means of geostatistical analyses. Herein, the application of this methodology to the periodic spatiotemporal variability of the resident population related to home-work mobility is investigated. To do so, additional tools are developed to extrapolate the real-world population from the crowding maps of provider's clients.

Dynamic exposure maps can be set in a more general call for a dynamic approach to flood risk assessment and management (Viglione et al., 2014). Actually, flood risk varies over time not only with regard to climate non-stationarities and urban

- development trends, since hydrological, economical, political, technological and social processes are also involved. For instance, effective campaigns devoted to increase people awareness towards flood risk, or to promote their capability to undertake effective water proofing practices, or to exploit warning systems, can dramatically diminish the flood risk over time. The same occurs by keeping the memory of flood disasters. These processes are inter-related and evolve over time (Di Baldassare et al., 2013). In this regard, some Authors proposed dynamic agent-based models to assess the temporal change in
- flood risk (Dawson et al. 2011; Haer et al., 2016). Such models are capable to perform a spatially-distributed analysis of flood risk, accounting for multiple factors, or agents, and their action-feedback relationships.

It must be pointed out that dynamic risk maps should account for the spatial heterogeneity of urbanization, in order to obtain precise assessments. Flood prone urbanizations can feature very different characteristics relevant to both exposure and vulnerability, even inside the same watershed. Land use is the first discrimination level to be considered, as people densities

- and temporal patterns could significantly differ in commercial, industrial, service, transport and residential areas. Secondly, the fabric type may also have remarkable impacts on the overall risk. Indeed, Fuchs et al. (2015) evidenced significantly different exposures among various types of land use (tourist accommodation, commercial, recreational, residential) in Austria. Urbanization heterogeneity is also relevant for flood risk studies in developing countries as shown by Vu and Ranzi (2017); in their assessment of flood risk in central Vietnam, they estimated the exposure and vulnerability of building and people by
- collecting questionnaires including data on building types.

To demonstrate the potentials of the geostatistical analysis herein proposed, it is applied to a suitable case study, identified in the western outskirts of Brescia (Lombardy, northern Italy). A detailed knowledge of the flooding dynamics and a sizeable set of mobile phone data are available for this watershed. The suggested approach made it possible to derive reliable dynamic exposure maps with respect to the land coverage and the calendar time periods, obtaining estimates of the expected number of

120 people affected by flood hazards along with its uncertainty. Hence, the paper is organized according to the following sections: (*i*) firstly, the innovative aspects of the geostatistical analysis methodology herein utilized are illustrated, (*ii*) secondly, the main hydraulic-hydrologic features of the analysed study area are described along with the available mobile phone data, (*iii*) the methodology application and the results are finally discussed.

## 2 Analysis methodology

The proposed geostatistical approach relies on Erlang mobile phone measures. An Erlang is the unit of measure of traffic intensity in a telecommunication system or network and it is widely used to quantify load and efficiency. The name is a tribute to A. K. Erlang (1878-1929), a Danish mathematician and statistician who firstly worked on traffic engineering (Erlang, 1909). In this study, Erlang measures consist in two-dimensional matrices which provide the spatial distribution of the average number of mobile phone users (MPU) bearing a SIM connected to the network, within a temporal interval and inside a spatial region.

- These data are collected by mobile phone providers and recorded at constant time steps with reference to a georeferenced grid of square cells. The availability of such a kind of data is progressively capturing the attention of urban planners (Becker et al. 2011; Calabrese et al., 2015), as they offer a variety of potential applications. In this study, the MPU spatiotemporal variability was summarized by means of daily density profiles (*DDP*), that provide the variability within a day of the MPU referred to a spatial region of interest. Such regions are inundation areas, thus expressing the spatiotemporal variability of people exposed
- to the flood risk. To define *DDP*, let  $e_{it}$  be the number of MPU in the *i*-th grid cell in a generic time interval *t*. Let  $I_r = \{i_1, ..., i_m\}$  be the set of grid cells in region *r* of interest. Furthermore, let define  $T_d = \{t_1, ..., t_o\}$  be the set of intervals of time in a day *d*. The daily density profile (*DDP*<sub>rd</sub>) can be defined according to Eq. (1), as a vector of the sums of MPU (a sum for each considered time instant) in region *r* and day *d* (length *o*):

$$DDP_{rd} = \{\sum_{l=1}^{m} e_{il,t1}, \sum_{l=1}^{m} e_{il,t2}, \dots, \sum_{l=1}^{m} e_{il,t0}\}'.$$
(1)

- Herein, the interest lies in analysing and classifying the occurrences in a time series of  $DDP_{rd}$ , related to a set  $d = \{d_1, ..., d_n\}$ of *n* analysed days. More precisely, the proposed approach firstly involves the clustering of similar  $DDP_{rd}$ , as discussed in detail in the following Section 2.1. The clustering procedure consists of two steps. In the first one, MPU spatial variability inside region *r* is considered by changing index *i* in a  $R^2 x$ -*y* coordinate space; to do so a data reduction strategy is applied. In the second one, the  $DDP_{rd}$  temporal variability is evaluated by changing index *t* in a  $R^1$  space.
- The characteristics of our data raise some issues related to the choice of the clustering technique to be chosen. In fact, traditional techniques (Arabie and De Soete, 1996) may not produce robust results when the number of variables is larger than the number of observations. Our data amount to *n* observations and p = m\*o variables (number of MPU per day). For instance, let us consider a case in which available data refer to one year (i.e. n = 365): information in each cell of the grid is available 4 times per hour (thus, o = 96) and the region is covered by 500 grid cells (m = 500). It follows that the number of variables is much
- larger than the number of observations (p > n) and so we refer to a high-dimensional data setup (Donoho, 2000). In highdimensionality some issues need to be considered, such as those of the curse of dimensionality (Keogh and Mueen, 2017). Bouveyron et al. (2007) addressed this issue with regard to clustering. However, as suggested by Jovi et al. (2015), a suitable solution is represented by data reduction. To do so, the Histogram of Oriented Gradients (HOG) approach is used in this paper. Therefore, data reduction works on index *i*, in order to convert the support from  $R^2$  (*x-y* coordinate space) to  $R^1$ .
- Once the  $DDP_{rd}$  are clustered in statistically similar groups, the total number of people in set  $T_d$  and region r can be estimated and associated with descriptive bands (DB), as discussed in Section 2.2. In this regard, there is crucial concern due the lack of MPU data from all companies providing phone services in northern Italy. To deal with this problem, as firstly suggested by Metulini and Carpita (2020), the approach proposed in this paper adopts a strategy to infer the total number of people by matching census data to available mobile phone data.

#### 160 **2.1 Data reduction and clustering**

To cluster similar *DDP* a technique for high-dimensional data reduction is adopted first. Then, reduced data are analysed by using a high-dimensional data clustering. Separately for each element of set  $T_d$  (i.e. for a given *t*), let  $\varepsilon_{it} = \{e_{1,t}, e_{2,t}, \dots, e_{im,t}\}$ 

be the MPU vector of region *r* in time instant *t* (dimension *m*). The aim is to reduce  $\varepsilon_{it}$  to the vector of values  $\kappa_{it}$  (dimension m' 
- sum of squares (*Tot*  $SS_H$ ) for different values of *H* that needs to be minimized with respect to *H*. For a certain *H*, the total within sum of squares is defined as *Tot within*  $SS_H = \sum_{i=1}^{H} W_{SS}(C_i) = \sum_{i=1}^{H} \sum_{k_d \in C_i} (\kappa_d \mu_i)^2$ , where  $\mu_i$  is the centroid vector (length  $S^*k^*o$ ) for cluster *i*; the total sum of squares is defined as *Tot*  $SS_H = \sum_d (\kappa_d \mu)^2$ , where  $\mu$  is the centroid vector for the full set of data. At this point the elements in  $d = \{d_1, ..., d_n\}$  (the days) have been assigned to a number of clusters  $C_1, ..., C_H (C_i \cap C_j = \emptyset, \forall i = 1, ..., H and \forall j = 1, ..., H with i \neq j$ ).
- In the second step, when the MPU variability over time is accounted for, we consider the vector  $DDP_{rd}$  (for a given region *r*) as the collection of functional observations  $x_{rd}$  ( $T_d$ ),  $T_d \subseteq (t_1, ..., t_o)$  of length *o*, with *d* varying in  $d = \{d_1, ..., d_n\}$  (i.e.  $\sum_{l=1}^{m} e_{il,t1}$  in  $t_l$ ). To do so, we adopt a model-based functional data clustering method (Becker et al., 2011) since it is more flexible than the alternatives: to each cluster it provides an estimated functional curve with specific parameters. We group days *d* (cluster's objects) in terms of the *o*  $DDP_{rd}$  values (cluster's variables), separately for each cluster of the previous step. Similarities in the
- functional form of the  $DDP_{rd}$  are considered, if viewed in terms of curves of values (y-axis) with respect to time instants (x-

axis). The curves of each group are modelled by using a group-specific set of distributional parameters (see Becker et al., 2011 for details). The adopted method can be used in high-dimensionality, since the clustering process employs the criterion of the sub-space clustering (Agrawal et al., 1998) which is adopted when one is only interested in considering the minimum number of variables needed for grouping objects, thus to reduce the dimensionality. In detail, it is herein proposed to adopt the

200 following path: i) functional data outlier detection by likelihood ratio test (LRT) is adopted to remove the anomalous  $DDP_{rd}$ , as proposed by Febrero-Bande et al. (2008); ii) the Bouveyron et al. (2015) clustering method is applied, using *funFEM* package in *R*.

The aim of this strategy is to assign the elements in  $d = \{d_1, ..., d_n\}$  (the days) to final clusters  $C_{F,I}, ..., C_{F,Z}$  ( $C_{F,i} \cap C_{F,j} = \emptyset$ ,  $\forall i = 1, ..., Z, \forall j = 1, ..., Z$ ), with  $Z \ge H$ . Thus, adopting these steps makes it possible to represent the dynamic of the MPU's

presences, in terms of a representative DDP for each group of days. Representative here means that each group includes days that are similar, in terms of index *i* (spatial distribution of MPU) and index *t* (temporal dynamic of MPU), and dissimilar from those included in other groups.

## 2.2 Population assessment

With the clustering strategy it is possible to display the amount of mobile phone users in region r for a set of time instants in

- clusters of days. However, the estimate of the total amount of people is needed for developing dynamic exposure maps. Unfortunately, data availability regards only one mobile phone company. To have a reliable estimation of the total number of people actually present in the study area, users of other mobile phone providers must be considered, as well. Collecting all these data is either unfeasible or unsustainably expensive. To perform analysis on a national scale, a convenient solution is represented by the use of the market share of the provider company, that can be applied to "correct" the *DDP*<sub>rd</sub>. Hence, an
- estimation of the total number of people can be obtained (e.g. let  $s_n$  to be the national level share assuming values in the range [0,1], the correct *DDP* would be  $DDP_{correct} = \frac{DDP_{rd}}{s_n}$ ). Country-level estimates are available from *Il Sole 24 Ore* newspaper (Il sole 24 ore, 2017). However, the market share usually varies significantly among cities, according to different social-economic characteristics of users. For instance, per-capita revenues are on average 19,514 €/year in Italy and 23,418 €/year in the Brescia Municipality (data by Ministry of Economy and Finance, Department of Finance, 2016), whereas the percentages of foreigners
- are 8.5 % and 18.5 %, respectively (data by Italian National Institute of Statistics ISTAT, 2017). Furthermore, families featuring more than 4 people are about 21.0 % in Italy and 16 % in the Brescia Municipality, while the percentage of people older than 65 is quite near to the national average of about 22.0 % (ISTAT, 2017).
  Thus, to guitable estimate the mericat share, the smallest level of correspondent propresented by the "Serien idi Consimente" (SC).

Thus, to suitably estimate the market share, the smallest level of aggregation, represented by the "Sezioni di Censimento" (SC) (i.e. population census districts), was used in this study (ISTAT, 2017). The following strategy is suggested: the number of

residents from administrative archives is compared with the number of TIM users in a residential area in the late evening hours. Bering in mind the characteristics of the social dynamics of the analysed residential areas, it is reasonable to assume that, in the late evening hours, residential SCs are only populated by residents. The comparisons, using data from ISTAT (*Anagrafe Comunale*), were performed separately for each SC.

Since the MPU grid is made of square cells while SCs are irregular polygons, the number of TIM users belonging to each SC was estimated by intersecting these spatial data. Thus, the portion of the cell belonging to the SC polygon was calculated in order to count how many TIM users are present in each polygon, by using the function *extract* in *raster* package, *R*. Let *Cell<sub>j</sub>*;  $j = 1, 2, ..., J_{SC}$  be the TIM cells (pixels) which overlap a chosen SC, the ratio  $A_j$  in Eq. (2)

$$A_j = \frac{area(SC) \cap area(Cell_j)}{area(Cell_j)},\tag{2}$$

represents how much of *Cell<sub>j</sub>* is included in the chosen SC (for each SC,  $A_j > 0$ ;  $j = 1, 2, ..., J_{SC}$ ). If *Cell<sub>j</sub>* is completely covered by SC, then  $A_j = 1$ , otherwise  $A_j < 1$ . Let *TUC<sub>j</sub>* be the density of TIM Users in *Cell<sub>j</sub>*, the estimation of the number of TIM users in SC *ETU<sub>SC</sub>* is computed as shown in Eq. (3).

$$ETU_{SC} = \sum_{i} TUC_{i} * A_{i} \tag{3}$$

The Estimated TIM Market Share in SC *ETMS<sub>SC</sub>* is thus given by the ratio in Eq. (4), where  $P_{SC}$  is the resident number assessed by population census for the SC.

$$240 \quad ETMS_{SC} = \frac{ETU_{SC}}{P_{SC}} \tag{4}$$

Differently from  $s_n$ , the  $ETMS_{SC}$  range is not necessarily in [0,1], since  $P_{SC}$  could be smaller than  $ETU_{SC}$ . An application example of this procedure can be found in Metulini and Carpita (2019b). In the count of  $P_{SC}$ , children (younger than 11 years) and elderly people (older than 80 years) were excluded, aiming at taking into consideration only people bearing a smartphone. The distribution of  $ETMS_{SC}$  can be used as a proxy for the TIM market share on a city level. More specifically, it appears to

be convenient to use the median of the distribution of  $ETMS_{SC}$ . The median is preferable to the mean in those cases when the distribution is symmetric. Metulini and Carpita (2019b) showed the presence of a strongly asymmetric distribution of  $TMS_{SC}$  for the case study of the city of Brescia. Moreover, Carpita (2019) showed good results in terms of the comparison between estimated people and official data by using the median for the case study of the Lake of Iseo during the Floating Piers. Let me(.) be the median statistics, the estimate  $DDP_{rd}$  for a given region r for a given day d is finally given by Eq. (5).

$$250 \quad \widehat{DDP}_{rd} = \frac{DDP_{rd}}{me(ETMS_{SC})} \tag{5}$$

#### 2.3 Result representation

For the sake of result interpretation, a graphical representation is herein adopted. Let us consider the  $DDP_{rd}$  vector to be a functional curve  $x_{rd}$  ( $T_d$ ) displaying, in the *y*-axis, the sum of MPU in region *r* and day *d* with respect to time instants  $T_d \in (t_l, ..., t_o)$  in the *x*-axis. Functional box plots (FBP) (Sun and Genton, 2011; 2012) can be used to display the profile for representative days.

In FBP a cluster of curves is ordered in term of its "band depth". A "median" value and an "envelope" are generally used to define the functional counterpart of traditional descriptive bands. Moreover, one can detect outlier curves. Specifically, a curve is an outlier if it exceeds the margins of the envelope by 1.5 in at least one considered time instant.

Separately for each final cluster, a FBP is derived. Let us consider cluster *h*, let  $d_h = \{d_{1h}, ..., d_{nh}\}$  be the group of days that 260 belong to cluster *h*, and let  $\widehat{DDP}_{rd,h} = [\widehat{DDP}_{r,d1h}, ..., \widehat{DDP}_{r,dnh}]$  be the matrix of dimension  $o*n_h$  with a  $\widehat{DDP}_{rd}$  in each column. By considering each vector as a curve, the FBP representing the profile plot of the total number of people (that we will call "city users", or simply "users") at different hours (with DB) on representative days is computed using matrix  $\widehat{DDP}_{rd,h}$ .

#### **3** Case study description

- The study area was selected as an emblematic and widespread situation of the unacceptable high flood risk affecting the foothill zone of the Po River Valley (Lombardy Region, northern Italy). It lies on the western outskirts of Brescia (Figure 1) and is overall included in the watershed of the Oglio River, a primary left-bank tributary of the Po River. The main drainage is supplied by the Mella River, bounding the eastern side of the study area, that is a left-bank tributary of the Oglio River. As can be seen in Figure 1, the study area features five natural streams originating from the southern boundary of the Alpine chain. From West to East, they are: Laorna, Gandovere, Vaila, La Canale and Solda.
- Before the area was anthropized, most of such streams had probably flooded the alluvial plain swamping into marshes, without a main outlet in the main river network. This was the result of both their almost ephemeral regime and the endorheic morphology of their watersheds. As the agricultural use of the alluvial plain grew, these streams were connected to the constructed irrigation-drainage network, in order to exploit their low flow for irrigation purposes and to drain the flood flow into the Mella River. These constructed downstream reaches feature two main drainage canals, the Gandoverello canal and the
- Mandolossa canal, depicted in Figure 1. They have become the artificial downstream reaches of the five watersheds, providing a drainage capacity in the South direction both for the mountain watersheds and low lands located in the alluvial plain. The total catchment area amounts to 112.3 km<sup>2</sup>, featured by an average imperviousness of about 22%; further details on the watershed hydrologic characteristics are available in the supplementary material.

In particular, the Laorna drainage path was strongly manipulated by a constructed straight canal 7 kilometres long, that diverts

- its streamflow towards the Gandovere stream and intercepts additional surface runoff produced by the northward low land. Both streams thus confluence into the Gandoverello canal. This was formerly the downstream reach of the Gandovere stream, whose limited conveyance capacity made it necessary to decrease the hydrologic load. Thus, a flow divider was constructed upstream of the Laorna confluence, so that almost half of the Gandovere streamflow is diverted towards the Mandolossa canal inlet by means of a diversion constructed canal that intercepts the Vaila stream. All these flow discharges, along with those coming from the La Canale stream and the Solda stream, converge into the inlet of the Mandolossa canal, which is
- characterized by the largest conveyance capacity in the study area.

This drainage network is also exploited by the irrigation system of Franciacorta, a vineyard-agricultural district located West of the study area, for the final disposal of the residual flow discharges. In Figure 1 two of the most important irrigation canals belonging to this system, Seriola Castrina and Seriola Nuova, are reported. Their discharges mainly affect the Gandovere and

290 Laorna streamflows. Further contributions come from the West Brescia outskirt, in terms of both urban stormwaters and irrigation excess flows, which directly drains into the Mandolossa canal. Finally, water table resurgences of the high Po River Valley are also present downstream of the Mandolossa canal inlet. Their fresh waters are however intercepted and drained by the downstream reach of the Mandolossa canal.

### 3.1 Hazard mapping

- Since the late 50s of the last century, this area has been subject to a deep urbanization sprawl, yielding the present land cover condition depicted in Figure 2. The dramatic increase in both the urban fabric and the industrial-commercial coverages has occurred at the expenses of croplands and permanent crops, so that sparse and isolated fabrics have evolved in a continuous and heterogeneous urbanization. In addition, various transport units have been upgraded to speedways and two highways were constructed. The flood risk perception in this this area has historically been related to the Mella River inundations, which
- affected the Brescia outskirts in the late 60s of the last century, until its riverbed underwent important engineering works. Conversely, with reference to the secondary stream network, the absence of a clear risk perception allowed such an urbanization sprawl to occur regardless of the floodplain extents. As Figure 2 clearly shows, most of the urban fabric areas and the industrial and commercial settlements are adjacent to the stream network.
- The increase in the land coverage yields huge impermeability degrees for the plain watersheds. Except for a combined sewer 305 overflow located in the West Brescia watershed that discharges into the Mella River, all the stormwaters produced by these urbanizations discharge into this secondary stream network, as well as those produced by many settlements in the Franciacorta district, which improperly exploits the irrigation system as a final receipt of combined sewer systems. Moreover, the low risk perception has led to a significant impairment of the functionality of these canals. The stream flow is now constrained into a number of narrow culverts and bridges with low decks and large piers. In addition, urbanized canals are no longer maintained,
- so that the riparian vegetation grows in an uncontrolled manner. The combination of the increase in the peak flow discharges and in the exposure, along with the decrease in the stream network conveyance capacity has led to a dramatic increase in the flood risk. Flood episodes explained by the secondary hydrographic network insufficiency have been observed since the late 90s, evidencing an empirical frequency of occurrence far less than 20 years. The hazard mapping was thus referred to return periods spanning from 5 years to 20 years, which are significantly less than those conventionally required in Italy for a
- secondary stream network to be considered verified (20-50 years).

The hazard analysis was herein conducted by using a design event method. As demonstrated by Balistrocchi et al. (2013) in this climatic context, a design event method is capable of providing results comparable to those of more sophisticated continuous approaches, if it is based on the Chicago synthetic hyetograph with a duration equal to the double of the catchment time of concentration. A classical leaf hydrologic model was developed in accordance with the sub-catchment subdivision

- illustrated in Figure 1. Extensive surveys of the stream cross sections and the inline structures were carried out to assess the actual conveyance capacities of the stream reaches. Flooding volumes were hence estimated by limiting the flood hydrographs generated through the hydrologic model to the overflow threshold discharges. Surveys of the historical flooding extensions that occurred during the last three decades addressed the delimitation of the flood prone areas. The total amount of exposed urban areas is about 160 ha (return period 5 yr), 230 ha (return period 10 yr) and 330 ha (return period 20 yr). Most of such
- areas are devoted to industrial-commercial settlements (70%), whereas the remaining part includes residential areas of medium-low density featuring similar fabric types (detached or semi-detached houses). Distinguishing between these two classes was found to be useful in decreasing the uncertainty of people estimates. The resulting flood hazard map is reported in Figure 2 along with the land cover, and it highlights the large amount of residential fabrics, industrial and commercial settlements potentially affected by flood events featuring low-medium return periods. An unacceptably high flood risk is
- therefore evidenced for the study area. Most of such areas were too dispersed or small to have suitable intersections with MPU grid cells. Therefore, they were grouped into four macro-areas referred to the individual canals and the land use, that was classified in urban fabrics (red) and commercial-industrial settlements (dark green). As shown in Figure 2 they are: the Laorna and Gandovere streams confluence (areas 1 and 5), the La Canale and Solda streams (areas 2 and 6), the southern Gandovere canal (areas 3 and 7), the Mandolossa canal in Roncadelle Municipality (areas 4 and 8).

#### 335 **3.2** Available mobile phone data

In this work we use mobile phone data provided by Telecom Italia Mobile (TIM), that is currently the largest mobile phone operator in Italy. According to the national economic newspaper, TIM's national share amounted to 30.2 % in December 2016 (*Il Sole 24 ore*). In our analysis, Erlang measure data represent the average number of both calling and non-calling mobile phone SIMs (assigned to a certain cell of the considered grid in a certain quarter. Statistical research in the area of urban

- planning using Erlang measure data is increasing: for example Carpita and Simonetto (2014) and Metulini and Carpita (2019a) studied the dynamics of people's presence at big events in the city of Brescia; Zanini et al. (2016) applied an Independent Component Analysis (ICA) method, for separating the city of Milan in a few main areas. Other works (Manfredini et al., 2015, Secchi et al., 2015) used Erlang measure data to study the dynamics of people's presence in Milan.
- In this study, reliable Erlang measures of MPU recorded by the TIM company are available. The investigated area is marked in a black solid line in Figure 1 (WGS 84 UTM 32 N coordinates: 5,040,920–5,049,980N, 585,970–592,970E, area about 64 km<sup>2</sup>) and is centred on the Mandolossa-Gandovere network. The area is covered by a georeferenced grid of square cells with 150 m sides, which provides the number of TIM users every 15 minutes. More precisely, for each cell of the grid and for each interval of time, the recorded data refer to an average measure of the number of mobile phones simultaneously connected to the network. For instance, Figure 3 shows a detail of the spatiotemporal distribution of TIM MPUs occurred on Wednesday,
- November 18<sup>th</sup> 2015, in a sample area of 20x20=400 cells, near the Mandolossa inlet. Therein, exposed areas, obtained by intersecting the urban covers with the flooding areas, were also reported. Thus, Figure 3 provides a sequence of snapshots of a dynamic map of people exposure to floods. As can be seen, the spatial distribution of raw data is realistic, as major densities

are observed along the main street network and in the urban areas. The temporal variability is also reasonable; for instance, lower densities are evidenced during nighttime in industrial sectors and main streets; see, for instance, the industrial settlement

near the confluence of the La Canale stream in the Mandolossa canal (flooding area marked with 6 in Figure 2). In these data, the information about the user mobility is hidden, meaning that one cannot trace the path followed by a single MPU over time. Measures are available for the period 2014–2016, even though after data inspection a more limited subset was found to be suitable for the analysis (from July 1<sup>st</sup> 2015 to August 11<sup>th</sup> 2016), due to data collection issues.

#### 4 Analysis procedure application

## 360 4.1 Procedure parameterization

The application of the HOG procedure to reduce the dataset dimensionality was performed for each quarter of a day in  $T_d$ , by dividing the full grid in 9 smaller grids  $G_i$ , i = 1, ..., 9. The parameter  $\sqrt{S}$  was thus set at 3. For each  $G_i$ , gradients and direction were then computed and the histogram of oriented gradients choosing k=4 bins corresponding, respectively, to angles 0°–45°,  $45^{\circ}$ –90°, 90°–135° and 135°–180° was obtained. In general, one can improve the recognition of the analysed grid by increasing

- the number of bins. This value was chosen in order to maximize *k* but, at the same time, to avoid the presence of HOG features with zero values. In each quarter, the extracted features are  $3^{2*4} = 36$ . Therefore, the order of dimensionality reduction is of  $400/36 \approx 11$ . The final vector  $\kappa_d$  for the sample area near the Mandolossa inlet (sample area evidenced in Figure 3), with all the quarters stacked for the same day and for day *d*, contains 36\*96 = 3456 features.
- The hierarchical *k*-means cluster analysis, with days used as the objects of the cluster and the features of  $\kappa_d$  used as cluster variables, was performed on a total amount of 360 days (d = 360) from July 1st 2015 to August 11th 2016. After data inspection, only the days of the last available year (from July 1st 2015 to August 11th 2016) were included in the analysis, since the first year (April, 2014 to June 30th 2015) features some collection problems. In effect, a configuration with 3 clusters sharply separating the days of the first year (till June 2015) and the days of the second year (by July 2015) was estimated, by performing the cluster analysis by using the full set of data. In the final sample all holidays were removed, in consideration of their specific
- characteristics with respect to typical days. More precisely, August, 15<sup>th</sup>, 1<sup>st</sup> and 2<sup>nd</sup> November, 8<sup>th</sup> December, 24<sup>th</sup> to 26<sup>th</sup> December, 31<sup>th</sup> December, January 1<sup>st</sup> and January 6<sup>th</sup>. 27<sup>th</sup> and 28<sup>th</sup> March (Easter), 25<sup>th</sup> April, May 1<sup>st</sup>, June 2<sup>nd</sup> were removed. In addition, those days where a large amount of data (>10%) were missing were removed, as well. Conversely, data in those days where missing data were less than 10% were maintained. A test for the possible presence of curse of dimensionality, based on the distribution of the distances among pairs of objects, was performed. A unimodal distribution that suggests the
- absence of such a problem was derived. On the whole, the amount of suitable data appears to be sufficient to get reliable estimates of people exposed to the flooding risk in the study area.

The number of first-step clusters was chosen according to the relative decreasing trend of the total within sum of squares with the group number increase. Figure 4 shows this trend, evidencing that a splitting in 4 clusters would decrease the total within

sum of squares by half with respect to a 1 cluster splitting. Since this decrease appears to be satisfactory, the sample of days

- was split in H = 4 clusters, where,  $C_1$  corresponds to all the days mostly occurring in July, August and September (green spineplots shown in Figure 5),  $C_2$  corresponds to working days mostly occurring from February to June (blue spine-plots shown in Figure 5),  $C_3$  corresponds to working days mostly occurring from October to January (red spine-plots shown in Figure 5) and  $C_4$  corresponds to the weekends except for those included in cluster  $C_1$  (yellow spine-plots shown in Figure 5).
- Hence, MPU variability over time instants was accounted for by considering the Mandolossa's *DDP* of each day as a functional curve. Firstly, those days that have to be considered outliers were removed by using the curve outlier detection method (Febrero-Bande et al., 2008) separately for each first-step cluster. Secondly, it was evaluated whether days should be further grouped in terms of dissimilarity in the *DDP* functional curve dynamic. To do so, the assumption of independence of our functional data was tested by using Portmanteau (Gabrys and Kokoszka, 2007) and distance correlation (Székely and Rizzo, 2013) tests. Model based functional data clustering techniques (Bouveyron et al., 2015) suggests splitting the "summer" group
- in 3 sub-groups, containing, respectively, the days of July, the days of August and the days of September. This second-step splitting leads to Z=6 final clusters, where  $C_{F,1}$  includes days of July,  $C_{F,2}$  includes days of August,  $C_{F,3}$  includes days of September and  $C_{F,4}$ ,  $C_{F,5}$  and  $C_{F,6}$  match, respectively,  $C_2$ ,  $C_3$  and  $C_4$ .

Illustrations of *DDPs* for representative days by using functional box plots are reported in Figure 6 (residential fabrics) and in Figure 7 (commercial and industrial settlements), for each of the 8 flooding areas shown in Figure 2 for the 10 year return

- period. Although functional data clustering suggests splitting into six groups, for the sake of clarity, the summer months (July, August and September) were combined in a single final cluster (Cluster 1, C1), as well as all the working-days from October to June in a single final group (Cluster 2, C2) and the weekends from October to June (Cluster 3, C3), thus leading to 3 clusters. To extract the number of people in each quarter of each day from the grid's cells to the irregular polygon of each area (i.e. to find  $DDP_{ri,d}$  for area  $r_i$ , i = 1, ..., 8, and day *d*) the procedure described in Section 2.2 was applied. Hence, MPUs were firstly
- divided by a constant c = 0.85, in order to consider children (>12 years) and old people (>80 years), who likely do not have smartphones (i.e. about 85% of the people are in the age range [12,80] in Brescia). Then, by estimating the median value of the market share ratio at the SC level by adopting the methodological strategy in Section 2.2, which amounts to about 20%, the estimated  $DDP_s$  for each area and for each day were derived by applying Eq. (5). The estimated market share is also consistent with that found by Carpita and Fabbris (2019). Estimated  $DDP_{ri,d}$  underwent the functional box plot strategy,
- separately for days d in the 3 clusters (with outliers excluded) and for  $r_i$  corresponding to the 8 areas illustrated in Figure 2.

## 4.2 Results and discussion

Figure 6 and Figure 7 show, respectively, the resulting functional box plots for residential and productive areas, where the estimated number of city users in different time instants is indicated. The three clusters of days are reported separately. Overall, the number of city users is lower during the first hours of the day and increases in the morning, reaching a peak during working

415 hours (9 am–1 pm and 2–6 pm), both in residential and in productive areas. In the Moie di Sotto residential area located at the confluence of the Laorna stream and the Gandovere stream (flooding area 1 in Figure 2), the people number is estimated at

about 200, during the first hours of the day and during the night and increases to about 250 during working hours (inhabitant density is about 25–30 ha<sup>-1</sup>). The dynamic, similar in all the three clusters, shows irrelevant differences among different periods of the year.

- In the Villaggio Badia residential area located North of the Mandolossa canal inlet (flooding area 2 in Figure 2), the city user number varies between a minimum of 1200 people and a maximum of 1400 people, during an average day (inhabitant density is about 30–35 ha<sup>-1</sup>). During the working-days of the months from October to June (cluster C2), the peak reaches 1600 users. Moreover, the descriptive bands appear to be wider in summer (cluster C1) and on weekends (cluster C3) as compared to cluster 2, where bands are narrower (*i.e.* lower variability between days).
- Residential areas along the southern Gandovere canal (flooding area 3 in Figure 2) are not very populated. Only 50–70 users are there during an average summer day (cluster C1) or during the weekend (cluster C3) (inhabitant density is about 18–23 ha<sup>-1</sup>). The number amounts to more than 80 people on working hours of working days (cluster C2). The number of city users in Roncadelle's residential area located along the Mandolossa canal (flooding area 7 in Figure 2) is less sensitive to working hours, especially during summer. In summer, city users vary from a minimum of about 600 up to a maximum of 700. There
- are about 800 city users during working hours of working days and weekends (days belonging to clusters C2 and C3). Industrial and commercial settlements of Moie di Sotto (flooding area 5 in Figure 2) feature 1000–1500 people (night, first hours of the day–working hours) in summer. These numbers increase up to about 1200–1800 on working days and to about 1100–1600 at weekends. This high density is mainly due to the presence of a commercial outlet of regional interest in this area. Industrial and commercial settlements along the La Canale and Solda stream network (flooding area 6 in Figure 2) are
- very highly populated by city users and the difference between the number of people in summer and on working days is significant. Daily minimum and maximum values are included between 2000 and 3000 in summer, and between 2500 and 3500 during working days. Weekends follow a stable dynamic (about 2500 people throughout the entire day).

Flooding areas related to the southern Gandovere canal (flooding area 8 in Figure 2) presents a productive area with an average number of users varying from 250 to 380 in summer and from 300 to 420 during working days. In the same way as Villaggio

Badia (southern part of flooding area 2 in Figure 2), the number of users during the weekends is stable and it stands at about 320/330. The industrial sector of Roncadelle (flooding area 4 in Figure 2) is another highly populated area, featuring more than 2000 people during the day. On some particular working days and weekends this number reaches about 3000 (red dashed lines of outliers in Figure 7). In this area, a remarkable difference in the number of users between working days, summer days and weekends was not detected.

#### 445 **5** Conclusions

In this paper a novel approach to the assessment of the risk related to people exposure to floodings based on a geostatistical analysis of Erlang measures was proposed. Such a procedure takes advantage of data reduction (histogram of oriented gradients discussed in Section 2.1), in order to face the dimensionality curse issue. Its suitability and potentials were demonstrated with

regard to an urban outskirt area located near Brescia (Lombardy, Northern Italy), which is affected by widespread and frequent

- floodings. In Figure 3 the possibility of expressing the spatiotemporal variability of exposed people by using time variable maps was illustrated. These data feature high spatial resolution (150 m) and short time step (15'), thus providing reliable assessments even for the smallest analysed areas (about 4–5 ha) and a precise evaluation of the temporal dynamics. Indeed, daily density profiles can be derived according to this procedure. Then, these profiles can be clustered yielding groups of similar daily time patterns. Clustering results are definitively meaningful, since working days and weekends are acknowledged
- to show different temporal dynamics, when they belong to working months (from October to June). Conversely, daily dynamics in summer months (July, August and September, usually exploited for the longest holidays in Italy), must be regarded as different from the others. In addition, working days and weekends feature more similar daily density profiles during such months. As can be seen in Figure 6 and in Figure 7, the daily temporal variability of people exposed to floodings can be assessed with respect to the day cluster and the type of urban areas (residential or industrial–commercial), both in terms of
- expected value (the median) and uncertainty (confidence band), thus providing comprehensive information to agencies and authorities devoted to the flood risk management.

The need to assess the entire population would theoretically require the gathering of a huge amount of datasets, from all the providers that operate in the area of interest. This issue would lead to a remarkable increase in the data collection cost and would be difficult to overcome. Nevertheless, census data make it possible to infer the total population from the users of a single provider, by means of local estimates of its market share, as discussed in Section 2.2.

It is worth underlining that this statistical support, along with the high spatiotemporal resolution and the reliability of the raw data, makes the proposed procedure particularly appealing in order to decrease the errors of exposed people estimates. Such a support is not provided by crowdsourcing techniques, which are based on voluntary data supplies and commonly rely on very limited datasets with respect to the amount of the exposed people. A second advantage that must not be disregarded lies in the

- possibility of exploiting dynamic exposure maps, or alternatively the clustered daily density profiles, by directly implementing them in emergency plans, regardless of the potential malfunctioning of the mobile phone connection during the flood episode. Conversely, crowdsourcing could be strongly compromised by the difficulties of connecting to the network during the emergency period. Indeed, dynamic exposure maps derived by mobile phone data have strong potentials to substantially improve emergency plans, so that real-time rescues, relief supplies and traffic management could be better addressed.
- Future developments of the geostatistical approach proposed in this paper could be addressed towards multiple items which deserve further in-depth analyses. First, complete mobile phone data feature more additional information than those available for this study. Actually, datasets include the collection of matrices of OD-Origin-Destination vectors in different seasons, days of the week and hour of the day, which are constructed by using the SIM identification numbers. Hence, it would be possible to track users down. By knowing the density of origin-destination vectors of the paths around critical traffic nodes, it would
- be possible to forecast potential critical conditions for mobility and better manage traffic in a more precise manner. Second, coupling traffic management decision support systems with real-time rainfall-runoff-flooding modelling is also a research perspective being considered. Presently, the exploitation of mobile phone data in real-time is problematic.

Nevertheless, in the future, a more common use of 5G and GPS technologies in mobile devices will facilitate the real-time assessments of people spatial distribution. From a prevention perspective, this could make the identification of preferential

- traffic flows possible, thus evidencing potential risks during inundation onsets or emergency situations. Alternative safe pathways could be identified and communicated to exposed people, in order to facilitate their evacuation. Third, it would be possible to profile the SIM users, while remaining anonymous and respecting their privacy. Users could be categorized, in order to isolate specific targets from the whole user set, and their behaviours could be statistically analysed
- separately from the others. Thus, a future development of the statistical matching procedure between mobile phone data and census data could use demographic and socio-economic information about the SC areas, for example the ISTAT
- ARCH.I.M.E.DE database (www.istat.it/it/archivio/190365). Since it is likely to assume heterogeneous behaviours of individuals, this database will be beneficial in future works in order to organize individuals in classes in terms of their age, gender, income or their job. In fact, different mobile phone companies have different costs, and this may affect the choice of different classes of individuals.
- Fourth, exposed people behaviours and habits can significantly change after hydro-climatic alarms or during flood events, since people would be aware of flood hazard and limit their exposure in flood prone areas. Mobile phone data make the identification of anomalous exposed people mobility during alarms possible, while the geostatistical approach herein proposed provides a tool to analyse whether, and how far, people behaviours are different from those of common days. Actually, a sample far larger than two years would be beneficial to make such statistics more reliable, since alarms involve a few days in
- a year. It is worth noting that in the analysed area the risk perception towards the secondary network is almost absent, as well as a capillary local warning system. Flood risk perception is mainly related to the primary hydrographic network (i.e. Mella River). Therefore, regarding the specific test case, in this research the possibility of drastic changes in human behaviour during heavy rainfall alarms are not expected. Indeed, people virtuous behaviours are usually the result of extensive campaigns to raise public awareness against flood risk, coupled with trusted and effective warning systems. Hence, a final objective of the
- ongoing research will take into consideration the effectiveness of the non-structure practices that will be adopted to mitigate flood risk in the test watershed.

#### 6 Data availability

Mobile phone data can be shared after motivated request, due to restrictions enforced by the provider. Geographical data can

be freely downloaded at <u>http://www.geoportale.regione.lombardia.it/</u>. Hydrological simulation outcomes and models cannot be shared at this time, because they are part of ongoing research.

## 7 Author contributions

RR and MC delineated the research topic, guided its development and supervised the paper writing. MB and RM performed 515 the hydrologic analyses and the geostatistical analyses, respectively, and wrote the paper.

#### **8** Competing interests

The authors declare that they have no conflict of interest.

## 9 Acknowledgments

The Authors want to thank the two anonymous Reviewers and the scientific comment Author for their fruitful suggestions and their contributions to the paper improvement.

#### **10 Financial support**

This research was developed at University of Brescia in the course of the FLOod RIsk MAnagement Policies (FLORIMAP) Project which received funding by the Cariplo Foundation under grant number 2017/0708. The content of this article does not reflect the official opinion of the Cariplo Foundation. Responsibility for the information and views expressed in this paper lies

entirely with the authors. The research will continue after additional funding by the Lombardy Region (Mo.So.Re. project on sustainable and resilient mobility).

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
