# Peer review of "Dynamic maps of people exposure to floods based on mobile phone data"

_Natural Hazards and Earth System Sciences, 2020_

## Short Comment (SC1) · 15 Jul 2020

The article presents an interesting approach to overcoming a traditional flood risk framework that does not consider dynamic flood exposure maps, depending on the dynamics of urban life. The methodology presented allows to consider the spatial-temporal distribution of human activities that exposes people to different types of flood hazards, generating different responses to a flood disaster. This can lead to an effective emergency plan, flexible in accordance with exposure that varies spatially and temporally (during the day and throughout the year, as highlighted by the article's results), which can better manage the few available resources (e.g. a conscious use of rescue teams, targeted evacuation plans, etc). The use of mobile phone data allows to overcome the limitations deriving from the exposure estimates based simply

on night time population density or official statistical inventories data, widely used as a measure of exposure to flood hazards in much research, and could have a potential in helping governments and flood risk management authorities in assessing risk, issuing warnings, and planning emergency responses to urban natural disasters. However, if the methodology proposed correctly identifies human positions and people exposure, it does not take into account human disaster response behavior (during floods daily activities can drastically change). Some modeling techniques, including agent-based models (ABMs), have been recently introduced to the field of flood risk assessment to simulate the dynamic distribution of the population during flooding, while still introducing inevitable simplifications of the human behavior patterns and disaster responses. For an integrated flood risk management in the future, it will be increasingly essential to consider the feedback between floods and people in a dynamic way and I suggest to give a comment on this issue. A minor comment: at lines 285-289 and 297-299 there are little mistakes.

---

## Referee Comment (RC1) · Anonymous Referee #1 · 16 Sep 2020

The manuscript describes a method to analyse mobile phone user data for the aim of elaborating dynamic maps of population exposure to floods. The presented method of geospatial analysis is innovative and is interesting for future application in the field of flood risk analyses and flood risk management. The manuscript is well written and the conclusions base on the data and on the results. Thus, I suggest the publication of the manuscript after a few minor corrections. Abstract: The last sentence mentions the application of the method in real-time rescues and reliefs. However, this has not been demonstrated and the application in real-time lacks of a real-time data access. Moreover, real-time hazard maps (dynamic flood maps) are not available yet. Please discuss this need for dynamic hazard and exposure data and its accessibility in real-time in the discussion or conclusion section. Line 105: Please explain the term "Erlang

mobile phone measures" or give a reference to it. Figure 5: The figure is hard to understand. Figure 5a is referred in the figure caption as days a week but shows the months. Moreover, e.g., green color is refereed to days from July to September but the figure shows green also in October. The same is for blue and yellow. Please revise the figure accordingly or add an additional explanation. Lines 1-61 can be shortened remarkably.

---

## Referee Comment (RC2) · Anonymous Referee #2 · 7 Oct 2020

In this paper, the Authors propose an interesting mapping approach to the assessment of the risk related to people exposure to floods, based on a geostatistical analysis of Erlang measures. After an introduction of the main features of floods (starting from key factors such as urbanization and climate changes, to the components, i.e. hazard, vulnerability and exposure), the Authors focus on operational aspects of empirical investigations developed in this framework. First, data sources availability: mainly distinguishing between official and crowdsource data, the paper highlights the potentialities of mobile phone data in identifying, tracing and classifying human localization. Second, dimensions and dimensionality: to provide a good managerial and recommending instrument to local, regional and national policymakers, the ability of data to provide a comprehensive and detailed overview over time and space of individuals, becomes fundamental. On the other hand, datasets with such high detail of information correspond to a huge amount of data. As well depicted by Authors, these two aspects motivate the need of a statistically robust procedure to provide reliable results about dynamic mapping of citizens. Focusing on the interesting case-study of Brescia (Italy), the Authors classify and map individuals from July 1st 2015 to August 11th 2016, using mobile phone data. The procedure can be seen as a three-steps approach, where the first two are represented by dimensionality reduction steps via clustering, and the third is given by statistical matching of collected mobile information to census data, aiming at estimating the total amount of people in a specific area. The clustering procedure, together with the matching methodology used and proposed by Carpita and Metulini (2020), provide the possibility of exploiting exposure maps to flood risk, and specifically the temporal dynamics of exposed residents.

The originality of the data and the proposed techniques can represent a substantial contribution to the understanding of natural hazards and their consequences. The manuscript is well written, and the conclusions well summarize the main findings of the paper. Thus, I suggest the publication of the manuscript after a few minor reviews. Minor comments for the publication of the manuscript concern the methodology. In section 2.1, the Authors refer to a specific parameter: "k is a parameter that need to be chosen". One reference for the choice of k should be reported, especially for non-statistician readers. Moreover, the motivation for the choice of Bouveyron and Come (2015)'s procedure, among all possible functional data techniques, should be (briefly) addressed. Concerning the Carpita and Metulini (2020)'s statistical matching approach, are there any kinds of test, procedure, etc., to evaluate representativeness and reliability of the final result of the population assessment step? This aspect should be (briefly) addressed.

Further developments: Due to the richness of mobile information and the heterogenous moving behaviour of individuals during the day, several further developments can be considered for future works. For example, it would be very interesting (from
a prevention perspective) to restrict the sample of investigation and focus on the intraday mapping of individuals in meaningful time periods of the year, e.g. the months with highest probability of observing floods. Also the idea of considering the movement response of residents to floods may be strongly interesting. Similarly, possible insights may be evaluated for the statistical matching procedure in future works. For example, what happens sharing population in different classes? Assuming heterogeneity in the behaviour of individuals, can you include in the procedure the propensities of different classes of residents to the use of smartphones during the day?

Please also note the supplement to this comment:
https://nhess.copernicus.org/preprints/nhess-2020-201/nhess-2020-201-RC2-supplement.pdf

---

## Author Response (AR1)

**Authors' replies to Editor comments**

5

20

25

all)."

Q1: "while it is – correctly – reported that increasing risk has at least two root causes (effects and impacts of CC as well as population dynamics in flood-prone areas) I kindly would like to suggest that you add one or two statements on the latter. Also in Northern Italy we do have distinct regions with decreasing population, and some others with population increase, which makes the general conclusion that values in flood-prone areas increase impossible (this is valid for many areas, but not for

- A1: We thank the Editor for this comment, which gives us the opportunity for a broader discussion on this crucial issue. We agree with the Editor in identifying the demographic growth as the most important cause for the urbanization sprawl, so that in the regions where emigration prevails, less marked urbanization growth rates are observed. Thus, on a global scale, the urbanization growth rates feature a strong regional variability. However, a fundamental driver for urban sprawl must be considered in addition to population growth in developed countries, such as Italy. Indeed, the touristic demand for accommodations (hotels, holiday homes) leisure facilities and shopping centers leads to significant soil consumption rates,
- 15 even in those regions (Alpine and Apennine mountain valley, major islands, and Southern Italy) where resident population continues to emigrate towards other Italian regions or foreign countries, since several decades. Examples of population decline associated with high urbanization growth rates have been presented in the manuscript for the Italian case (lines 33-44) and two supporting references have been added.

CENSIS: 53° Rapporto sulla situazione sociale del Paese, Franco Angeli, https://www.censis.it/rapporto-annuale/sintesi-del-53%C2%B0-rapporto-censis, 2019. (in Italian)

ISPRA: Soil consumption in Italy - 2019 Edition, Report SNPA, 08/19, ISBN 978-88-448-0964-5, 2019. (in Italian)

Q2: "Moreover, as recently shown for Austria and Switzerland (Fuchs et al., 2015; Fuchs et al., 2017; Röthlisberger et al., 2017), elements at risk exposed to flood hazards are not always/not necessarily of the same building type and therefore exposure is highly dynamic in space and over time. To give an example, for Austria, it has been shown that a relatively higher share of buildings exposed to flood hazards in the lowlands belongs to the category of commercial buildings, while in the upper catchments this share is higher for residential buildings and hotels/buildings of the tourism industry. Consequently, if we include "population dynamics" or "dynamics in exposure" into a somehow dynamic risk concept, this should be mentioned. Obviously this does not necessarily mean that I expect a reference to my own works, this is only for illustration – there are

30 also other works centered on these issues available."

A2: We strongly agree with comment. In our paper we actually distinguished between industrial-commercial areas and residential areas, in order to decrease the uncertainty of exposed people estimates. A further classification refine was not found to be necessary, in view of the homogeneity of fabrics in these two classes. The need to incorporate spatial heterogeneity of

- 35 urbanization in dynamic risk mapping has been further underlined in the introduction section (lines 107-115) and further details on land use classification have been added in section 3.1 (hazard mapping) (lines 326-327) The following references which well suit the discourse have been added as examples of studies which taken into account the urbanization heterogeneity with reference to flood risk estimation.
- Fuchs, S., Keiler, M., and Zischg, A.: A spatiotemporal multi-hazard exposure assessment based on property data, Natural Hazards and Earth System Sciences, 15, 2127-2142, doi:10.5194/nhess-15-2127-2015, 2015.
   Vu, T. T., and Ranzi R.: Flood risk assessment and coping capacity of floods in central Vietnam. Journal of Hydro-Environment Research, 14, 44-60. doi:10.1016/j.jher.2016.06.001, 2017.
- 45 Q3: "Moreover, some of the items described in section 2 (methods) have overlaps with a paper by Metulini and Carpita (2020). As this concerns mainly the methods section, I personally can accept it (and also according to the overall NHESS guidelines this is somehow ok) but I kindly would like to ask you to change wording a bit so it is not 1:1 the same text."

A3: Sentences in method sections (in particular section 2) have been rephrased by keeping the same content but by using other 50 words. Moreover, the whole paper has been revised by a native English speaker. Authors' replies to short comment SC1 by Monego M.

55

80

85

We thank you for your general evaluation of the work and for the question that you arose, which gives us the opportunity of a deeper discussion on this aspect.

O1: "However, if the methodology proposed correctly identifies human positions and people exposure, it does not take into account human disaster response behavior (during floods daily activities can drastically change). Some modeling techniques, including agent-based models (ABMs), have been recently introduced to the in the field of flood risk assessment to simulate

- the dynamic distribution of the population during flooding, while still introducing inevitable simplifications of the human 60 behavior patterns and disaster responses. For an integrated flood risk management in the future, it will be increasingly essential to consider the feedback between floods and people in a dynamic way and I suggest to give a comment on this issue." A1: Indeed, exposed people behaviors and habits can significantly change after hydro-climatic alarms or during flood event occurrences, as well as their ability to decrease vulnerability by implementing flood proofing practices. However, these
- virtuous behaviors are usually the result of extensive campaigns to raise public awareness against flood risk, coupled with 65 trusted and effective warning systems. In the analyzed area the risk perception towards the secondary network is almost absent, as well as a capillary local warning system. In addition, the knowledge and a widespread application of flood proofing practices do not exist (both structural and non-structural). Flood risk perception is mainly related to the primary hydrographic network (i.e. Mella River). Despite the dramatic increase in flooding episodes and in consequent economic damages, the impairment
- 70 of the conveyance capacity of the hydrographic network and the urbanization sprawl still continue. Therefore, in the regards of the specific test case, in this research the possibility of drastic changes in human behavior during heavy rainfalls are not expected.

On the other hand, a dynamic approach to the flood risk is becoming mandatory, especially in consideration of the auspicatedfuture application of non-structural practices to the risk mitigation. Actually, agent-based modelling falls into the framework

- 75 of a dynamic assessment of flood risk. The methodology herein proposed has potential to monitor people mobility dynamic during crises, evidencing modifications of their spatiotemporal distribution. Mobile phone network hardly fails during floods, thus observations of people dynamic under crisis conditions could be beneficial for a better calibration of any dynamic model. Finally, mobile phone data are richer than the ones used in this study, since vector data allows provider to follow users along their path. This type of information was not available for this study. However, there are potentials to further improve this technique in order to assess mobility preferential ways and to change them to increase escape security.
- The need for a dynamic approach to flood risk assessment, along with the following references to ABMs, has been remarked in the introduction section, to better set this work inside the most updated research (lines 98-106). Then, a brief discussion on people behavior change during flood alarms has been added in the final part of the conclusion section.

Dawson, R. J., Peppe, R., & Wang, M. (2011). An agent-based model for risk-based flood incident management. Natural Hazards, 59(1), 167-189. doi:10.1007/s11069-011-9745-4

Haer, T., Botzen, W. J. W., & Aerts, J. C. J. H. (2016). The effectiveness of flood risk communication strategies and the influence of social networks-insights from an agent-based model. Environmental Science and Policy, 60, 44-52. doi:10.1016/j.envsci.2016.03.006.

4

90 Q2: "Minor comments: at lines 285-289 and 297-299 there are little mistakes"

A2: Repetition will be removed from section 3.1 (lines 285-289 of the first submission).

Authors' replies to referee comment RC1 by Anonymous Referee #1

**95**

We thank Referee #1 for her/his general evaluation of the paper and her/his supportive comments. All requested revisions has been implemented in the revised paper. Individual replies are listed below.

**Q1: "Abstract: The last sentence mentions the application of the method in real-time rescues and reliefs. However, this has not 100 been demonstrated and the application in real-time lacks of a real-time data access.**

A1: Actually the proposed exploitation of mobile phone data to estimate people exposed to floods is completely novel and it has not found verification during real-world episodes of inundations, yet. This issue will be remarked in the final sentence of the abstract (lines 18-20), that will be rewritten in order to make it clearer as follows: "This novel methodology still deserves verification during real-world flood episodes, even though it appears to be more reliable than crowdsourcing strategies, and seems to have potentials to better address real-time rescues and relief supplies".

105

**Q2: "Moreover, real-time hazard maps (dynamic flood maps) are not available yet. Please discuss this need for dynamic hazard and exposure data and its accessibility in realtime in the discussion or conclusion section."**

A2: A general need for a dynamic approach to flood risk assessment has been underlined by many authors, for instance those 110 who authored the publications reported in the references below. Reasons are manifold and are related to the dynamic behavior of decision makers, exposed people or rescuer actions, and to the dynamic nature of climate and urbanization. For instance, effective campaigns devoted to increase people awareness to flood risk or their capability to undertake water proofing practices, as well as, the implementation of waring systems can dramatically diminish the flood risk over time, without adopting structural strategies to decrease the flood hazard. Furthermore, accounting for urban development trends could be beneficial, in order to

115 assess the future increase in flood risk. A discussion on these aspects has been added in the introduction (lines 98-106) and the reference list will be improved with these updated references. Dawson, R. J., Peppe, R., & Wang, M. (2011). An agent-based model for risk-based flood incident management. Natural

Hazards, 59(1), 167-189. doi:10.1007/s11069-011-9745-4

Di Baldassarre, G., Viglione, A., Carr, G., Kuil, L., Salinas, J. L., & Blöschl, G. (2013). Socio-hydrology: Conceptualising human-flood interactions. Hydrology and Earth System Sciences, 17(8), 3295-3303. doi:10.5194/hess-17-3295-2013 120

Haer, T., Botzen, W. J. W., & Aerts, J. C. J. H. (2016). The effectiveness of flood risk communication strategies and the influence of social networks-insights from an agent-based model. Environmental Science and Policy, 60, 44-52. doi:10.1016/j.envsci.2016.03.006.

Viglione, A., Di Baldassarre, G., Brandimarte, L., Kuil, L., Carr, G., Salinas, J. L., . . . Blöschl, G. (2014). Insights from socio-125 hydrology modelling on dealing with flood risk - roles of collective memory, risk-taking attitude and trust. Journal of Hydrology, 518(PA), 71-82. doi:10.1016/j.jhydrol.2014.01.018

Moreover, the approach proposed in this paper does not need real-time data, since the exposed people assessments can be carried out of emergency periods. Once functional box plots of exposed people are derived, as those reported in Figure 6 and

- 130 in Figure 7, an assessment of people affected by floods can be expressed in terms of typical spatio-temporal patterns and their uncertainty, independently of the availability of real-time data. Thus the main aim of the paper is to pursue risk prevention by enhancing preparedness. The exploitation of mobile phone data in real-time will become more feasible after a widespread use of 5G and GPS technologies. A brief remark of this issue has been added to the final part of the conclusion section (lines 475-506). However, examples of almost real-time applications already exist, but in different contexts. For instance, Florence
- 135 municipality uses mobile phone data to assess tourist crowding and citizen mobility (daily, weekly and monthly). In general, mobile phone data accessibility is getting easier, as mobile phone providers are more conscious of their market value (an example is given by API developed by TIM, the largest provider in Italy, which allows stakeholders to download data in almost-real-time).
- 140 Q3: "Line 105: Please explain the term "Erlang mobile phone measures" or give a reference to it."

Raw data description at lines 125-131 has been improved as follows:

"The proposed geo-statistical approach relies on Erlang mobile phone measures. An Erlang is the unit of measure of traffic intensity in a telecommunication system or network and it is widely used to quantify load and efficiency. The name is a tribute to A. K. Erlang (1878-1929), a Danish mathematician and statistician who firstly worked on traffic engineering (Erlang, 1909).

145 In this study, Erlang measures consist in two-dimensional matrices which provide the spatial distribution of the average number of mobile phone users (MPU) bearing a SIM connected to the network, within a temporal interval and inside a spatial region. These data are collected by mobile phone providers and recorded at constant time steps with reference to a georeferenced grid of square cells."

**Erlang. A.K (1909). The theory of probabilities and telephone conversations. Nyt Tidsskrift Matematika, B.20, 33-39.**

150

Q4: "Figure 5: The figure is hard to understand. Figure 5a is referred in the figure caption as days a week but shows the months. Moreover, e.g., green color is referred to days from July to September but the figure shows green also in October. The same is for blue and yellow. Please revise the figure accordingly or add an additional explanation."

- A3: The caption of Figure 5 has been corrected and improved. References to panels in Figure 5 were inverted in the first
   version of the caption. The cluster names refer to the period that they mainly, but not exclusively, belong to. Thus, the green cluster indicates days mainly belonging to the period July-September, even though a few days of this cluster can be observed in October. Additional explanations of the spine plots have been added to section 4.1 (lines 384-388) and in the figure caption to precise this aspect.
- 160 Q5: "Lines 1-61 can be shortened remarkably."

Indeed, the first part of the introduction (lines 20-60) reports preliminaries which introduce known concepts. In our opinion this part is beneficial to place the work in a comprehensive-historical framework. On the other hand, considering the additional discussions required by reviews and comments, a better balance could be achieved by giving more room to the specific issues and research advances faced in the paper. Thus, we simplified this part by removing very well-known concepts and ancillary

165 discussions and by shortening some paragraphs. More precisely: lines 21-23: shortened; Lines 23-26: deleted; Lines 33-35: shortened; Lines 39-45: shortened; Lines 48-57: shortened; Line 58: deleted (references were deleted consistently). In the revised version, the preliminary part of the introduction involves lines 22-59, including the additional discussion flowing Editor's comments. This should be beneficial to shift the discussion focus towards the main aim of the paper and to better set the work in the most recent literature.

170

Authors' replies to referee comment RC2 by Anonymous Referee #2

We thank Referee #2 for her/his thorough reading and general evaluation of the paper and her/his suggestions for future developments. All requested revisions has been implemented in the revised paper. Replies are listed below.

175

**Q1: "In section 2.1, the Authors refer to a specific parameter: "k is a parameter that need to be chosen". One reference for the choice of k should be reported, especially for non-statistician readers."**

A1: As we have said in lines 337-338, the criterion used in our application aims to maximize k subject to avoiding the presence of zeros in the vector of HOG features (i.e. avoiding empty bins). According to Salhi et al. (2013), the larger the number, the

180 more accurate the results are. Moreover, according to them, parameter k usually ranges from 4 to 20 in the related literature. The text has been revised (lines 174-189).

We will add within the manuscript some details on it and the following reference: Salhi, A. I., Kardouchi, M., & Belacel, N. (2013). Histograms of fuzzy oriented gradients for face recognition. In 2013 International conference on computer applications technology (ICCAT) (pp. 1–5). IEEE

185

**Q2: "Moreover, the motivation for the choice of Bouveyron and Come (2015)'s procedure, among all possible functional data techniques, should be (briefly) addressed."**

A1: We have choosen Bouveyron and Come (2015)'s model-based functional data clustering method because its better flexibility compared to alternative methods. As already reported in lines 167–171 of the original version of manuscript, we

190 need a method in which to each cluster it corresponds an estimated functional curve with specific parameters. In fact, our aim is to consider the similarities in the functional form of the daily density profiles (DDPs), viewed as a curve of values (y-axis) with respect to time instants (x-axis). Adopting a model-based functional data clustering method, each group's curves are modelled by their own set of distributional parameters.

Moreover, since we have high dimensional dataset (more variables, or features, than observations), as we wrote in lines 171-

195 173, the chosen method is suitable for these special kinds of dataset, because it applies sub-space clustering (Agrawal et al., 1998) which is generally adopted to consider just the minimum number of variables needed for grouping objects, thus reducing the dimensionality.

In lights of the referee comment, we will rephrase the related part of the manuscript in order to be more clear and more exhaustive (lines 197-201).

200

Q3: "Concerning the Carpita and Metulini (2020)'s statistical matching approach, are there any kinds of test, procedure, etc., to evaluate representativeness and reliability of the final result of the population assessment step? This aspect should be (briefly) addressed."

A3: There are no specific tests proposed in the literature suitable to our case. Moreover, a comparison test using official data
is not possible since the lack of official data. However, the proposed procedure it is the result of a series of tests. In particular, we think that a key aspect related to the reliability of the population assessment step is the choice of the measure of central tendency used in the denominator of equation 5 (line 221). The choice of preferring the median instead of the mean is motivated by the strongly asymmetrical distribution of Estimated TIM Market Share (ETMS), as shown in Metulini, Carpita (2019b). The choice of the median has been then tested by Carpita (2019) for the case of the Lake Iseo during the Floating Piers (there official data are available). Results show that the estimated number of people is similar to that provided by official sources. Revised text at lines 338-343.

Q3: "Further developments: Due to the richness of mobile information and the heterogenous moving behaviour of individuals during the day, several further developments can be considered for future works. For example, it would be very interesting

215 (from a prevention perspective) to restrict the sample of investigation and focus on the intraday mapping of individuals in meaningful time periods of the year, e.g. the months with highest probability of observing floods. Also the idea of considering the movement response of residents to floods may be strongly interesting.

A3: The project has recently been refinanced by Lombardy Region (the local authority of the administrative region where the study catchment is placed) and a new partner is TIM (the most important mobile phone provider in Italy). This allows us to

- 220 have access to the full set of data that providers routinely collect. Datasets include a vector data reporting the individual user location along with their SIM identification number. On the one hand, it would be possible to track users down. As a perspective the ongoing research, just started, includes the collection of matrices of OD-Origin-Destination vectors in different seasons, days of the week and hour of the day. By knowing the density of vectors with origin and destination of the paths around critical traffic nodes it will be possible, more precisely, to forecast potential critical conditions for mobility and better
- 225 manage traffic. Also coupling traffic management decision support systems with real-time rainfall-runoff-flooding modelling is a research perspective being considered. From a prevention perspective, this could make it possible the identification of preferential traffic flows evidencing potential risks during inundation onsets or emergency situations. Alternative safe pathways could be identified and enforced to exposed people, in order to facilitate their evacuation. On the other hand, it would be possible to profile the SIM users, even though keeping anonymousness and respecting their privacy. Thus, users could be
- 230 categorized (with respect to age, gender, etc.) in order to isolate specific targets from the whole user set. Thus their behaviors could be statistically analyzed separately from the others. Unfortunately, such additional data were not available for this study, but in the forthcoming development these research advances could be addressed. Another issue regards people behaviors during flood emergency or warnings, which could significantly change with respect to

those of ordinary days. The geostatistic analysis proposed in our work is able to detect possible differences in the exposed

235 people spatial distribution, that are statistically significant. A further objective of the research extension is the implementation of a warning system in the study area, coupled with a campaign to make people aware of flood risk associated with the analyzed stream network. Mobile phone data will be useful to evidence the actual response of expose people to these non-structural practices and to estimate the expected decrease in the flood risk. To have a statically robust assessment, a larger set of data is however needed, since emergency or warning occurrences are rare (a few days in a year). The extension of the project will increase the dataset size and will make this research advances easier. As dataset size increase, it would be possible to restrict

240 increase the dataset size and will make this research advances easier. As dataset size increase, it would be possible to restr the analysis to specific periods (maybe the rainy seasons)A discussion on further developments has been added to the final part of the conclusion section (lines 475-506).

Q4: Similarly, possible insights may be evaluated for the statistical matching procedure in future works. For example, whathappens sharing population in different classes? Assuming heterogeneity in the behaviour of individuals, can you include in the procedure the propensities of different classes of residents to the use of smartphones during the day?

A future development of the statistical matching procedure between mobile phone data and census data could use demographic and socio-economic information about the SC (sezioni di censimento) areas, for example the ISTAT ARCH.I.M.E.DE database (www.istat.it/it/archivio/190365). Since, it is likely to assume heterogeneous behaviors of individuals, we may think to use

250 ARCH.I.M.E.DE. database in future works to share individuals in classes in terms of their age, gender, income or their job. In fact, different mobile phone companies have different costs, and this may affect differently the choice of different classes of individuals. This discussion has been added in the final part of the conclusion section (lines 475-506).

**Dynamic maps of people exposure to floods based on mobile phone**

**255 **data**

260

Matteo Balistrocchi1, Rodolfo Metulini2, Maurizio Carpita3, Roberto Ranzi4Ranzi4

1Department of Engineering "Enzo Ferrari", University of Modena and Reggio Emilia, Modena (MO), 41125, Italy

4Department of Civil, Environmental, Architectural Engineering and Mathematics, University of Brescia, Brescia (BS), 25123, Italy

2Department of Economics and Statistics, University of Salerno, Fisciano (SA), 84084, Italy
 3Department of Economics and Management, University of Brescia, Brescia (BS), 25122, Italy
 44Department of Civil, Environmental, Architectural Engineering and Mathematics, University of Brescia, Brescia (BS), 25123, Italy
 265

Correspondence to: Matteo Balistrocchi (matteo.balistrocchi@unimore.it)

Abstract. Floods are acknowledged as one of the most serious threats to human people's lives and properties worldwide. To mitigate the flood risk, it is possible to act separately on its components: hazard, vulnerability, exposure. Emergency
 management plans can actually provide effective non-structural practices to decrease both people exposure and vulnerability. Crowding maps depending on characteristic time patterns, herein referred to as dynamic exposure maps, provide represent a valuable tool to enhance the flood risk management plans. In this paper, the suitability of mobile phone data to derive crowding maps is discussed. A test case is provided by a strongly urbanized area subject to frequent floodings located in on the western outskirts of Brescia town (northern Italy). Characteristic exposure spatio-temporal patterns and their uncertainties were detected, with regard to land cover and calendar period. This novel methodology still deserves verification during real-world flood episodes, even though it appears to be more reliable than crowdsourcing strategies, and seems to have potentials to better address real-time rescues and reliefs suppliesyThis novel methodology appears to be more reliable than crowdsourcing strategies, and has potentials to better address real-time rescues and reliefs supply.

**1** Introduction**

280 Floods are natural phenomena whose hazards afflict nearly 20 million people worldwide (Kellens et al., 2013), posing a serious challenge to the protection of human people's lives and the liveability of urban settlements. Both low-income and high-income countries are strongly impacted by extreme weather events and a A high-confidence increase trend in the resulting economic damages and social costs due to extreme weather events has globally been documented (Kreibich et al., 2019). As reported by Munich RE (2020) over the period 1980-2019, flooding accounts for some 40% of all loss related natural catastrophes, with

ha formattato: Italiano (Italia)

ha formattato: Italiano (Italia)

ha formattato: Inglese (Regno Unito)

285 losses worldwide totalling more than US\$ 1tn. For instance, floods and landslides in Thailand in 2011 resulted in 43 US\$ bn, that is the highest flood losses of all time.

Two major factors can be advocated for justifying such a trend: climate change and increased urbanization and people exposure (Hartmann et al., 2013). These factors involve different components of the risk concept (UN ISDR, 2009), which is given by the combination of hazard, exposure and vulnerability. Climate change was popularly acknowledged as a leading cause for the

- 290 increases in the frequency and intensity of heavy storms and, consequently, of the flood episodes-hazard (Solomon et al., 2007). This factor is therefore related to the hazard component of the flood risk. However, according to the Intergovernmental Panel on Climate Change (IPCC) IPCC (Hartmann et al., 2013), and as confirmed also by up-to-date analyses of flood intensity in Europe (Blöschl et al., 2019), the absence of a global likely trend in the incidence of floods arises. RThis can be due to various reasons lie in: the high regional variability of heavy storm trends, in terms of type, magnitude and significance, as well as in
- 295 the strong influence played by the watershed hydrologic characteristics and the local flood management practices on the flood generation processes.

On the contrary, population urbanization represents a likely global trend, though characterized by a strong regional variability. MPeople migration from countryside or mountain areas to cities is the main driver of urban sprawl. In 2008, for the first time in human history, more than half the world population was living in urban settlements and the percentage continues to augment

- 300 (UN DESA Population Division, 2012). Touristic demand is an additional driver for urbanization growth, that plays a peculiar role in developed countries. For instance, in Italy many areas are affected by emigration, namely Alpine and Apennine valleys, southern regions and islands, and urbanization growth rates are equal to or greater than the national average rate. A clear example is provided by Southern Italy, where the annual rate of soil consumption between 2017 and 2018 was 0.23 %, largergreater than the national average national one-of 0.21 % (ISPRA, 2019), whileeven if Southern Italy faced a population
- 305 decrease of 1.5 % between 2015 and 2019 (the national average loss was estimated at 0.7 %) (CENSIS, 2019). An additional example is provided by the Sondrio province (in the mountainous part of the Adda River basin, Lombardy, Northern Italy), where the yearly soil consumption per capita is 1.11 m2, whereas the regional one is 0.63 m2. On the contrary, the population urbanization represents a non-climaticlikely global trend, though characterized by a strong

310 for the first time in human history, more than half the world population was living in urban settlements and the percentage continues to augment (UN DESA Population Division, 2012). \_Population urbanization determines dramatic increases in people exposure and vulnerability to floods, since most of recent urbanizations are developed disregarding the natural

- extension of floodplains. Therefore, urbanizations often lie in flood prone areas and local communities are not able to put in place effective flood defence practices. Touristic demand is an additional driver for urbanization growth in developed countries.
- 315 For instance, in Italy many areas affected by people emigration, namely Alpine and Apennine valleys, southern regions and islands, urbanization growth rates are equal to or greater than the national average rate. A clear example is provided by Apulia Region (Southern Italy), where the soil consumption was about 8.43% in 2017 and 2018 (larger than the national one, equal to 7.63%) (ISPRA, 2019), while Southern Italy faced a population decrease of 1.5% between 2015 and 2019 (CENSIS, 2019).

UPopulation urbanization determines dramatic increases in people exposure and vulnerability to floods in immigration areas,

- 320 since most of recent urbanizations lie in flood prone areas and local communities are not able to put in place effective flood defence practices in place. Moreover, urbanization usually leads to the impairment of the conveyance capacity of the stream network, so that flooding areas are basically larger than in the undeveloped condition. Thus, the urbanization sprawl consequently results in increased damage to communities, private properties and public infrastructures, and, which is defined as the product of exposure and vulnerability for a given hazard. Indeed, this second factor must be regarded as the main cause
- 325 for the likely increasing trend of the-flood risk (Barredo et al., 2009). A number of researches on flood risk changes under economic and population growth scenarios indicate that this contribution is at least equal to, but commonly larger than, the climate change one (Feyen et al., 2009; Maaskant et al., 2009; Bouwer et al., 2010; Te Linde et al., 2011; Rojas et al., 2013). This flood risk trend forced an unavoidable shift in the paradigms of flood defence, by recognizing that not all events can be completely controlled and that structural practices have limits (Johnson and Priest, 2008). For instance, the European Flood
- 330 Risk Management Directive (European Union, 2007) acknowledges that floods cannot be stopped from occurring, and that Thus, the focus must be placed on how to mitigate the damages to flood prone communities (European Union, 2007). OHence, over the last decades, flood risk management has evolved from a structural-based defence approach, aiming at decreasing the hazard component, towards a more holistic perspective (Merz et al., 2010; Arrighi et al., 2019), -taking into consideration drivers and impacts of flood risk. Vyulnerability and exposure were thus investigated in a deeper manner than
- 335 in the past, whilst nNovel concepts were introduced, such as residual risk (UN ISDR, 2009), accounting for the potential structural failure of the defence system (Vorogushin et al., 2009; Schumann, 2017; Balistrocchi et al., 2019), and resilience, that is the ability to recover from a damage or to absorb an impact (Liao, 2012).

Actually, vulnerability reduction plays an essential role for successful adaptation to flood risk (Kreibich et al., 2017). 
[revised manuscript text omitted]

- 405 The order to demonstrate the potentials of the geostatistic geostatistic analysis herein proposed, it is applied to Aa suitable case study, has been identified in the western outskirts of Brescia town (Lombardy, northern Italy), for which a A detailed knowledge of the flooding dynamics and a sizeable set of mobile phone data are available for this watershed. The suggested approach made it possible to derive reliable dynamic exposure maps with respect to the land coverage and the calendar time periods, obtaining estimates of the expected number of people affected by flood hazards along with its uncertainty.
- 410 Hence, the paper is organized according to the following sections: (*i*) firstly, the innovative aspects of the geostatisticgeostatistical analysis methodology herein utilized are illustrated, (*ii*) secondly, the main hydraulic-hydrologic features of the analysed study area are described along with the available mobile phone data, (*iii*) the methodology application and the results are finally discussed.

**2 Analysis methodology**

415 The proposed geo-statistical approach relies on Erlang mobile phone measures. An Erlang is the unit of measure of traffic intensity in a telecommunication system or network and it is widely used forto measuringquantify load and efficiency. The name is a tribute to A. K. Erlang (1878-1929), a Danish mathematician and statistician who firstly worked on traffic ha formattato: Tipo di carattere: (Predefinito) Times Nev Roman, Nessuna sottolineatura engineering (Erlang, 1909). In this study, Erlang measures consist in two-dimensional matrices which provides the spatial distribution of the average number of mobile phone users (MPU) bearing a SIM connected to the network, within a temporal

- 420 interval and inside a spatial region. These data are collected by mobile phone providers and recorded at constant time steps with reference to a georeferenced grid of square cells. The proposed geo-statistical approach relies on Erlang mobile phone measures, which consist in the average number of mobile phone users (MPU) bearing a connected SIM. These data are collected by mobile phone providers and recorded at constant time steps with reference to a georeferenced grid of square cells. The availability of such a kind of data is progressively capturing the attentionly arousing the enthusiasm of the urban planners2
- 425 community (Becker et al. 2011; Calabrese et al., 2015), as they offer a variety of potential applications. In this study, the MPU spatiotemporal variability was summarized by means of daily density profiles (*DDP*), that provides the variability within a day of the MPU referred to a spatial region of interest. Such regions are inundation areas, thus expressing the spatiotemporal variability of people exposed to the flood risk. To define *DDP*, let  $e_{it}$  be the number of MPU in the *i*-th grid cell in a generic time interval *t*. Let  $I_r = \{i_1, ..., i_m\}$  be the set of grid cells invelated to region *r* of interest. Furthermore, let define  $T_d = \{t_1, ..., t_m\}$
- 430  $t_o$ } be the set of time intervals of time in a day d. The daily density profile (*DDPrd*) can be defined according to Eq. (1), as a vector of the sums of MPU (a sum for each considered time instant) in region r and day d (length o-)-of values describing the sum of MPU in region r and day d:

 $DDP_{rd} = \{\sum_{l=1}^{m} e_{il,t1}, \sum_{l=1}^{m} e_{il,t2}, \dots, \sum_{l=1}^{m} e_{il,t0}\}'.$

(1)

Herein, the interest lies in analyzing analysing and classifying the occurrences in a time series of  $DDP_{rd}$ , related to a set  $d = \{d_1, ..., d_n\}$  of *n* analyzed analysed days. More precisely, the proposed approach firstly involves the clustering of similar  $DDP_{rd}$ , as discussed in detail in the following Section 2.1. The clustering procedure consists of two steps. In the first one, MPU spatial variability inside region *r* is considered by changing the-index *i* in a  $R^2 x$ -y coordinate space; to do so a data reduction strategy is applied. In the second one, the  $DDP_{rd}$  temporal variability is evaluated by changing index *t* in a  $R^1$  space.

The characteristicsClustering of our mobile phone data raise some issues related to the choice of the clustering technique to be 440 chosen. according to the above described procedure is not straightforward. In fact, tNevertheless, traditional techniques (Arabie and De Soete, 1996) may not produce robust results when the number of variables are larger than the number of observations. cannot be applied. In fact, Our ddata amount to *n* observations and p = m\*o variables (number of MPU informations pervalues each day). For instance, let us consider a case in which if one has one year of available data refered refer to one year (i.e. n =365): informations in each cell of the grid are variables 4 times per hour, repeated every 15 minutes (sothus, o = 96) and

- thein a region is covered by 500 grid cells (m = 500).5 It follows that the variable number of variables is wayfarmuch larger than the number of observations (p > n) and so we refer to an, depicting a high-dimensional data setup context. (Donoho, 2000). In When analyzing high-dimensionality data, someseveral issues need to be considered, such as those of the curse of dimensionality (Keogh and Mueen, 2017), need to be faced. With specific regard to data clustering, this issue has been addressed by Bouveyron et al. (2007) addressed this issue with regard to clustering. However, as suggested by Jovi et al. (2015), a suitable solution is represented by high dimensional data reduction provides a suitable solution. To do so, an the
- $430 (2013), \underline{a}$  suitable solution is represented by high dimensional data reduction provides a suitable solution. To do so,  $\frac{1}{410}$

approach based on the Histogram of Oriented Gradients (HOG) approach is used in this paper. Therefore, data reduction worksacts on index *i*, in order to convert the support from  $R^2$  (*x*-*y* coordinate space) to  $R^1$ .

Once the  $DDP_{rd}$  are clustered in statistically similar groups, the the total number of people in set  $T_d$  and in region r can be estimated and associated with descriptive bands (DB), as discussed in Section 2.2. In this regard, there is crucial concern is given bydue the lack of MPU data from all companies providing phone services in northern Italy. To deal with this problem, as firstly suggested by Metulini and Carpita (2020), the approach proposed in this paper adopts a strategy to infer the total number of people by matching census data to available mobile phone data to census data.

**2.1 Data reduction and clustering**

To cluster similar DDP a technique for high-dimensional data reduction is firstly adopted first. Then, reduced data are 460 analyzed analyzed by using a high-dimensional data clustering. Separately for each element of set  $T_d$  (i.e. for a given t), let  $\varepsilon_{it}$ = { $e_{1,t}, e_{2,t}, \dots, e_{im,t}$ }' be the vector of dimension *m* of MPU vector of *i*m region *r* in time instant  $t_{\underline{d}}$  (dimension *m*). The aim is to reduce  $\varepsilon_{it}$  to the vector of a new set of values  $\kappa_{it}$  (, of dimension m'

- parameter that is a parameter that needs to be chosen. The Larger is the parameter k is, the better are the results are; moreover, in related literature k usually ranges from 4 to 20 (Salhi et al., 2013). These matrices are used to derive the histogram of gradients with k equal bins, where k is a parameter that need to be chosen. The vector of features  $\kappa_{kl}$  is the vector of features given by all the elements of the derived betained histogram of oriented gradients, s when stacked  $\forall S$ . Considering that The length off
- 480 the elements of the derivedobtained histogram of oriented gradients, s when stacked  $\forall S$ . Considering that The length off vector  $\kappa_{it}$  is  $S^*k_{2}$ . Subsequently, the vector  $\kappa_{it}$  is stacked over the subscript *t*, in order to derive (for region *r* and day *d*) obtaining the vector of features  $\kappa_d$  for region *r* and day *d*, (of dimension  $S^*k^*o_2$ ).

ha formattato: Tipo di carattere: Non Corsivo, Inglese (Regno Unito)

ha formattato: Tipo di carattere: Non Corsivo, Inglese (Regno Unito)

ha formattato: Tipo di carattere: Corsivo ha formattato: Tipo di carattere: Corsivo

| ha formattato: Tipo di carattere: Corsivo, Inglese (Regno
Unito) |
|---------------------------------------------------------------------|
| ha formattato: Tipo di carattere: Corsivo, Inglese (Regno
Unito) |
| ha formattato: Tipo di carattere: Corsivo, Inglese (Regno
Unito) |
| ha formattato: Tipo di carattere: Corsivo, Inglese (Regno
Unito) |
| ha formattato: Tipo di carattere: Corsivo                           |
| ha formattato: Tipo di carattere: Non Corsivo                       |

 $\kappa_d$  is used contains the features s (variables) undergoing the in the first clustering step of the method, where days representsing which the objects to be clustered, in terms of, according to how the MPU are distributed distributed over region r according to index *i*, are represented by days. For all days in the data set  $d = \{d_1, \dots, d_n\}$ ,  $\kappa_d$  is computed for all the days in the data set  $d = \{d_1, \dots, d_n\}$ . 485  $\{d_1, \dots, d_n\}$ , and a k-mean cluster analysis (Hartigan and Wong, 1979), in which the objects to be clustered are the *p* days and  $\kappa_d$  contains the values of the  $S^*k^*o$  (with  $S^*k^*o < m^*o$ ) variables for day d to be attributed to a cluster, is used, is performed, where the *n* days are the objects to be clustered.  $\kappa_d$  contains the values of the  $S^*k^*o$  (with  $S^*k^*o < m^*o$ ) variables for day d to be attributed to a cluster. According to the Hartigan and Wong criterion, the number of clusters' number H is 490 chosen by looking to analyzing the decreasing trend of the ratio between the total within sum of squares (Tot within  $SS_{H}$ ) and the total sum of squares (Tot  $SS_H$ ) for different values of  $H_{\overline{z}}$  that needs to be minimized with respect to -the number of groups—H. For a certain H, the total within sum of squares is defined as Tot within  $SS_H = \sum_{i=1}^{H} W_{SS}(C_i) =$  $\sum_{i=1}^{H} \sum_{k_d \in C_i} (\kappa_d - \mu_i)^2$ , where  $\mu_i$  is the centroid vector (length  $S^*k^*o$ ) for cluster *i*; the total sum of squares is defined as Tot  $SS_H = \sum_d (\kappa_d - \mu)^2$ , where  $\mu$  is the centroid vector for the full set of data. At this point the elements in  $d = \{d_1, ..., d_n\}$ 495 (the days) have been assigned to a number of clusters  $C_1, ..., C_H (C_i \cap C_j = \emptyset, \forall i = 1, ..., H and \forall j = 1, ..., H with i \neq j)$ . In the second step, that in which To account for when the MPU variability over time is accounted for, we consider the, for a given region r, as objects, the vector  $DDP_{rdr}$  (for a given region r) considered as the collection of functional observations  $x_{rd}$  $(T_d), T_d \subseteq (t_1, ..., t_o)$  of length o, with d varying in  $d = \{d_1, ..., d_n\}$  (i.e.  $\sum_{l=1}^m e_{il,t_1} \text{ in } t_l$ ). To do so, we adopt a model-based functional data clustering method (Becker et al., 2011) since it is more flexible than the alternatives: to each cluster it provides 500 an estimated functional curve with specific parameters. We group, separately for each cluster of the previous step, days d (cluster's objects) in terms of the  $\rho$  DDPed values (cluster's variables), separately for each cluster of the previous step. We The aim atis to considering the sSimilarities in the functional form of the  $DDP_{ed}$  are considered, if viewed in terms of as a curves of values of values (y-axis) with respect to time instants (x-axis). In doing so, The curves of each groupeach groups) curves are modelled by using aby group-specifictheir own set of distributional parameters (see Becker et al., 2011 for details). 505 The adopted method we adopt can be usedt is suitable for in high-dimensionality-data, assince the clustering process employsapplies the criteriona of the sub-space clustering (Agrawal et al., 1998) which is adopted, adopted when one is only interested to in considering just the minimum number of variables-needed needed for grouping objects, thus to reduce the

d (cluster's objects) were grouped in terms of the *o* DDPrd values (cluster's variables), separately for each cluster defined in
 the previous step. The aim is to consider the similarities in the functional form of the DDPrd, seen as a curve of values (y-axis) with respect the x axis (time instants). In doing so, each group's curves are modelled by their own set of distributional parameters (see for details Bouveyron and Come, 2015). The method adopted in this work is suitable for high-dimensional dataset, since the clustering process applies the criteria of the sub-space clustering (Agrawal et al., 1998), adopted to consider just the minimum number of variables needed for grouping objects, thus reducing the dimensionality. In detail, it is herein
 proposed to adopt the following path: j)- functional data outlier detection by likelihood ratio test (LRT) is adopted to remove

dimensionality To do so, a model-based functional data clustering method (Bouveyron and Come, 2015) was adopted, and days

ha formattato: Tipo di carattere: Corsivo ha formattato: Tipo di carattere: Corsivo, Inglese (Regno Unito)

ha formattato: Tipo di carattere: Corsivo, Inglese (Regno Unito)

| ha formattato: Tipo di carattere: Non Corsivo                               |
|-----------------------------------------------------------------------------|
| ha formattato: Tipo di carattere: Corsivo, Inglese (Regno
Unito)         |
|                                                                             |
| ha formattato: Tipo di carattere: Corsivo, Inglese (Regno
Unito)  |
| ha formattato: Tipo di carattere: Corsivo, Inglese (Regno
Unito)         |
| ha formattato: Tipo di carattere: Corsivo, Inglese (Regno
Unito)         |
| ha formattato: Tipo di carattere: Corsivo, Inglese (Regno
Unito), Pedice |
| ha formattato: Tipo di carattere: Corsivo, Inglese (Regno
Unito)         |
| ha formattato: Tipo di carattere: Corsivo, Inglese (Regno
Unito), Pedice |
| ha formattato: Tipo di carattere: Corsivo, Inglese (Regno
Unito)         |
| ha formattato: Inglese (Regno Unito)                                        |
| ha formattato: Tipo di carattere: Corsivo, Inglese (Regno
Unito)         |
| ha formattato: Inglese (Regno Unito)                                        |
|                                                                             |

ha formattato: Tipo di carattere: Non Corsivo

the anomalous  $DDP_{rd}$  are removed using functional data outlier detection by likelihood ratio test (LRT), as proposed by Febrero-Bande et al. (2008); ji) the clustering method developed by Bouveyron et al. (2015) clustering method is applied, usingalong with funFEM package in R.

The aim of this strategy is to With this strategy we aim at assign the elements in  $d = \{d_1, ..., d_n\}$  (the days) to a number of final clusters  $C_{F,l}, ..., C_{F,Z}$  ( $C_{F,i} \cap C_{F,j} = \emptyset$ ,  $\forall i = 1, ..., Z$ ,  $\forall j = 1, ..., Z$ ), with  $Z \ge H$ . Thus, the adopting of these-two steps-would 520 permit-makes it possible to representa representation of the dynamic of the MPU's presences, in termsthe form of a representative DDP for each group of days in region r., where, with, with rRepresentative here we means intend that to each group belongs-includes days that are similar, in terms of index i (spatial distribution of MPU) and index t (temporal dynamic of MPU), each other similarar each other and and dissimilar in between, from those included in other groups, in terms of index 525

*i* (spatial distribution of MPU) and index *t* (temporal dynamic of MPU)

**2.2 Population assessment**

With the ft the clustering strategy makes it is possible to displayrepresent the dynamic of the amountpresence of mobile phone users in region r forever a set of time instants in clusters of daysfor a group of "representative" days, an additional strategy is needed However, -to the estimate of the total amount of people is needed for developing dynamic exposure maps. 530 IndeedUnfortunately, usuallyin most times, data-are availabilityle regards just foronly one mobile phone company. To have a reliable estimationed of the total number of people that are actually present in the study area, users of other mobile phone providers must be considered, as well. Collecting all these data is either unfeasible or an-unsustainably expensive. To perform an national level analysanalysis is at on a national levelscale, a convenient solution is represented by the use of the market share of the provider company, that can be applied to "correct" the DDPrd. Hence, an estimatione of the total number of people can 535 be obtained (e.g. let  $s_n$  to be the national level share assuming than can assume values in the range [0,1], the correct DDP wouldmay be  $DDP_{correct} = \frac{DDP_{rd}}{S_{rec}}$ ) can be obtained. Country-level estimates are available from through Il Sole 24 Ore newspaper (Il\_sole\_24\_ore, 2017). However, the market share usually varies significantly among cities, according to different socialeconomic characteristics of users. For instance, per-capita revenues are on average 19,514 €/year in Italy and 23,418 €/year in the Brescia Municipality (data by Ministry of Economy and Finance, Department of Finance, 2016), whereas the percentages of foreigners are 8.5 % and 18.5 %, respectively (data by Italian National Institute of Statistics ISTAT, 2017). Furthermore, 540 families featuring more than 4 people are about 21.0 % in Italy and 16 % in the Brescia Municipality, while the percentage of people older than 65 is quite near to the national average of about 22.0 % (ISTAT, 2017).

Thus, to suitably estimate the market share, the smallest level of aggregation, represented by the "Sezioni di Censimento" (SC) (i.e. population census districts), was used in this study (ISTAT, 2017). TWe suggedt the The following strategy is suggested 545 is suggested: we compare thedata on the number of residents from administrative archives is compared with were compared to the number of TIM users on in a residential area in the late evening hours. Bering in mind the characteristics of the social dynamics of the analysedis residential areas, it is reasonable to assume that, in the late evening hours, -during these hours

ha formattato: Tipo di carattere: Non Corsivo

ha formattato: Tipo di carattere: Corsivo, Inglese (Regno (Inito)

residential SCs are only populated only by residents. The Such comparisons, using data from ISTAT (Anagrafe Comunale), were performed separately for each SC. using ISTAT data (Anagrafe Comunale).

550 SFirst, since the MPU grid is made of square cells while SCs are irregular polygons, the number of TIM users belonging to each SC was estimated by intersecting these spatial data. Thus, to count the number of TIM users in each polygon-the portion of the cell belonging to the SC polygon were was calculated in order to count how many TIM users are present in each polygon, by using the function extract in raster package, R. Let Cell;  $j = 1, 2, ..., J_{SC}$  be the TIM cells (pixels) which overlaps overlapping a chosen SC, the ratio  $A_i$  in Eq. (2)

555
$$A_j = \frac{area(SC) \cap area(Cell_j)}{area(Cell_j)}$$

560

which represents how much of the portion of Cellj is includedeovered inby the chosen SC (for each SC,  $A_j > 0; j = 1, 2, ...,$  $J_{SC}$ ). If Celli is completely covered by in SC, then  $A_i = 1$ , otherwise  $A_i < 1$ . Let  $TUC_i$  be the density of TIM Users in  $Cell_i$ , the estimationed of the number of TIM users in SC  $ETU_{SC}$  is computed as shown in Eq. (3). (3)

$$ETU_{SC} = \sum_{j} TUC_{j} * A$$

The Estimated TIM Market Share in SC ETMSSC is thus given by the ratio in Eq. (4), where  $P_{SC}$  is the resident number assessed

(5)

(2)

- by the population census the for the SC.  $ETMS_{SC} = \frac{ETU_{SC}}{P_{SC}}$ (4)
- Differently from  $s_n$ , the ETMSSC range is not necessarily in [0,1], since  $P_{SC}$  ETUSC could be smaller than larger than ETUSC  $P_{SC}$ . An application example of this procedure can be found in Metulini and Carpita (2019b). In the count of  $P_{SC}$ , elderly people 565 (>80 years) and children (younger than <-11 years) and elderly people (older than 80 years) and were excluded, aiming at taking into consideration only people bearing a smartphone. The distribution of ETMSSC can be used as a proxy for the TIM market share at on a city level. More specifically, it appears to be convenient to use the median of the distribution of ETMSsc The median is preferable to the mean in those cases when the distribution is symmetric. Metulini and Carpita (2019b) showed the presence of a strongly asymmetric distribution of *TMSSC* for the case study of the city of Brescia. Moreover, Carpita (2019)
- 570 showed good results in terms of the comparison between estimated people and official data by using the median for the case study of the Lake of Iseo during the Floating Piers. More specifically, it appears to be convenient to use the median of the distribution of ETMSSC, which is preferable to the mean in those cases when the distribution is asymmetric. Let me(.) be the median statistics, the estimate  $\overline{DDP}_{rd}$  for a given region r for a given day d is finally given by Eq. (5).

$$\widehat{DDP}_{rd} = \frac{DDP_{rd}}{me(ETMS_{SC})}$$

**575 2.3 Result representation**

For the sake of result interpretation, a graphical representation is herein adopted. Let us consider the the vector  $DDP_{rd}$  vector to be a functional curve of functional observations  $x_{rd}$  ( $T_d$ ) displaying representing, in the y-axis, the sum of MPU in region r and day  $d_{-(in y axis)}$ -with respect to, in the x-axis, time instants  $T_d \in (t_1, ..., t_o)$  in the x-axis (in the x axis). Functional box

ha formattato: Tipo di carattere: Corsivo

ha formattato: Tipo di carattere: Corsivo, Inglese (Regno Unito) ha formattato: Tipo di carattere: Corsivo, Inglese (Regno Unito), Pedice ha formattato: Tipo di carattere: Corsivo, Inglese (Regno Unito)

ha formattato: Tipo di carattere: Corsivo, Inglese (Regno Unito), Pedice

ha formattato: Tipo di carattere: Corsivo

plots (FBP) (Sun and Genton, 2011; 2012) can be used to display tThe profile for representative days.-can be displayed by 580 using functional box plots (FBP), the analogue of the traditional box plot for curves (Sun and Genton, 2011; 2012).

- In FBP a cluster<del>group</del> of curves is ordered in term of itsusing the concept of "band depth". Aa "median" value and an "envelope" is generated, that is are generally can be used to define thea functional counterpartversion of traditional descriptive bands. Moreover, it is possible toone can assign a curve to the outlier group detect outlier curves. "Specifically, a curve is an outlier -if it exceeds by 1.5 the margins of the envelope margins by 1.5 in at least one considered time instant.
- 585 Separately for each final cluster, aA FBP strategy-is derived performed separately for each final cluster. Let us consider cluster  $h_{\perp}$ -and-let  $d_h = \{d_{1h}, \dots, d_{nh}\}$  be the group of days that belonging to cluster h, and let  $\overline{DDP}_{r,d,h} = [\overline{DDP}_{r,d,1h}, \dots, \overline{DDP}_{r,d,nh}]$  be the matrix of dimension  $o^*n_h$  with a  $\widehat{DDP}_{rd}$  in each column. Let By econsidering each vector as ato be a curve, the FBP representing the profile plot of the total number of people (that we will call "city users", or simply "users") in at different hours (with DB) inon representative days is applied computed using to matrix  $\widehat{DDP}_{rd,h}$  to generate the profile plot estimating the 590 dynamic of the total number of people (that we will call "city users", or simply "users") in different hours (with DB) in representative days.

**3 Case study description**

[revised manuscript text omitted]

in Italy for a secondary stream network to be considered verified (20-50 years). The urban areas exposed to floodings are
 estimated in 160 ha, 231 ha and 330 ha, with respect to 5 years, 10 years and 20 years return periods. About 30% of those areas
 are residential fabrics whereas the remaining 70% is given by industrial and commercial settlements.

The hazard analysis was herein conducted by using a design event method. As demonstrated by Balistrocchi et al. (2013) in this climatic context, a design event method is capable to of providinge results comparable to those of more sophisticated continuous approaches, if it is based on the Chicago synthetic hyetograph with a duration equal to the double of the catchment 650 time of concentration. A classical leaf hydrologic model was developed in accordance with the sub-catchments subdivision illustrated in Figure 1. Extensive surveys of the stream cross sections and the inline structures were carried out to assess the actual conveyance capacities of the stream reaches. Flooding volumes were hence estimated by limiting the flood hydrographs generated through the hydrologic model to the overflow threshold discharges. Surveys of the historical flooding extensions that occurred during the last three decades addressed the delimitation of the flood prone areas. The total amount of exposed 655 urban areas amount tois about 160 ha (return period 5 yr), 230 ha (return period 10 yr) and 330 ha (return period 20 yr). Most of such areas are devoted to industrial-commercial settlements (70-%), whereas the remaining part is includes residential fabrics areas of medium-low density featuring similar fabric types (detached or semi-detached houses). To decrease the uncertainty of people estimates, dDistinguishing between these two classes was found to be useful in decreasing the uncertainty of people estimates. The resulting flood hazard map is reported in Figure 2 along with the land cover, and it highlights the large amount 660 of residential fabrics, industrial and commercial settlements potentially affected by storm-flood events featuring low-medium return periods. An unacceptably high flood risk is therefore evidenced for the study area. Most of such areas were too dispersed or small to have suitable intersections with MPU grid cells. Therefore, they were grouped into four macro-areas referred to the river networkindividual canals and the land use, that was distinguished classified inbetween urban fabrics (red) and commercial-industrial settlements (dark green). As shown in Figure 2 they are: the Laorna and Gandovere streams confluence 665 (areas 1 and 5), the La Canale and Solda streams (areas 2 and 6), the southern Gandovere canal (areas 3 and 7), the Mandolossa

canal in Roncadelle Municipality (areas 4 and 8).

**3.2 Available mobile phone data**

In t2This work we use focuses on mobile phone data provided by Telecom Italia Mobile (TIM), thatwhich is currently the largest mobile phone operator in Italy. in this sector. According to the national economic newspaper, TIM's national share amounted to 30.2 % in December 2016 (*Il Sole 24 ore*). In our analysis, Erlang measure data represent the average number of both calling and non-t-calling mobile phone SIMs (<del>both calling and not calling</del>) that are-assigned to a certain thatcell of the considered grid grid's cell in a certainthat quarter. Statistical research in the area of urban planning using Erlang measure data is increasing: for example have already been used in the context of urban planning along with statistical methods by Carpita and Simonetto (2014) and Metulini and Carpita (2019a), who studiedanalyzed the dynamics of people's presence inof people duringat big events in the city of Brescia; by Zanini et al. (2016), who find, by appliedyingmean of an Independent Component Analysis (ICA) method, for separatinge the city of Milan in a few main areas. a number of spatial components that separate main areas

of the city of Milan, and by oOther works (Manfredini et al., 2015, Secchi et al., 2015) used Erlang measure data to study the dynamics of people's presence in Milan.

In this study, reliable Erlang measures of MPU recorded by the TIM company are available. The investigated area is marked in a black solid line in Figure 1 (WGS 84 UTM 32 N coordinates: 5,040,920-5,049,980N, 585,970-592,970E, area about 64 680 km2) and is centered centred on the Mandolossa-Gandovere network. The area is covered by a georeferenced grid of square cells with 150 m sides, which provides the number of TIM users every 15 minutes. In details More precisely, for each grid's cell of the grid and for each time interval of time, the corresponding recorded data refers to anthe average measurenumber of the number of mobile phones simultaneously connected to the network. For instance, Figure 3 shows a detail of -the 685 spatiotemporal distribution of TIM MPUs occurred on Wednesday, November 18th 2015, in a sample area of 20x20=400 cells, near to the Mandolossa inlet. Therein, exposed areas, obtained by intersecting the urban covers with the flooding areas, were also reported. Thus, Figure 3 provides a sequence of snapshots of a dynamic map of people exposure to floods. As can be seen, the spatial distribution of raw data is realistic, as major densities suitably concentrate are observed along the main street network and in the urban areas. The temporal variability is also reasonable; for instance, lower densities are evidenced during nighttime 690 in industrial sectors and main streets; see, for instance, the industrial settlement near to-the confluence of the La Canale stream in the Mandolossa canal (flooding area marked with 6 in Figure 2). The mobility feature In of these data, the information about the user mobility is hidden, meaning that one can not cannot it is not possible to trace the path followed by a single MPU over time. Measures are available in-for the period 2014–2016, even though after data inspection a more limited subset was found to be suitable for the analysis (from July 1st 2015 to August 11th 2016), due to data collection issues.

**695 4 Analysis procedure application**

**4.1 Procedure parameterization**

The application of the HOG procedure to reduce the dataset dimensionality was performed for each quarter of a day in  $T_{d_s}$  by dividing the fulloriginal grid in 9 smaller grids  $G_i$ , i = 1, ..., 9. The parameter  $\sqrt{S}$  was thus set at 3. For each  $G_i$ , gradients and direction were then computed and the histogram of oriented gradients choosing with k=4 bins corresponding, respectively, to

- angles 0°\_45°, 45°\_90°, 90°\_135° and 135°\_180° was obtained. In general, one can improve the recognition of the analyzedanalysed gridobject is improved by increasing the number of bins. This value was chosen in order to maximize *k* but, at the same time, to avoidavoiding the presence of HOG features with a zero values of zerozeros amongin the vector of HOG features. In each quarter, the eExtracted features arecount for 32\*4 = 36 in each quarter, so Therefore, which correspond to a the order of dimensionality reduction is of n the order of 400/36 ≈ 11. The final vector  $\kappa_d$  for the sample area near to the
- 705 Mandolossa inlet (sample area evidenced in Figure 3), with stacked all the all-quarters stacked of for the same dayy stacked for sample area near to the Mandolossa inlet (sample area evidenced in Figure 3) and for day *d*-, countsamounts tocontains 36\*96 = 3456 features.

ha formattato: Tipo di carattere: Corsivo

The hierarchical *k*-means cluster analysis, with days used hereas the objects of the cluster are the days *d* and the features of  $\kappa_d$ , used as cluster variables, are represented by the features of  $\kappa_d$ , was performed on a total amount of 360 days (*d* = 360) from

[revised manuscript text omitted]

Residential areas along the southern Gandovere canal (flooding area 3 in Figure 2) are little-not very populated. Only 50–70 users are there during an average summer day of summer (cluster C1) or during the week-end (cluster C3) (inhabitant density

is about 18-23 ha-1). The number amounts to more than 80 people on working hours of working days (cluster C2). The nNumber of city users in Roncadelle's residential area located along the Mandolossa canal (flooding area 7 in Figure 2) is less sensitive to working hours, especially during summer. In summer, the number of city users variyes from a minimum of about 600 up to a maximum of 700. City usersThere are about 800 city users during working hours of working days and weekends (days belonging to clusters C2 and C3).

ha formattato: Tipo di carattere: Non Corsivo ha formattato: Tipo di carattere: Non Corsivo ha formattato: Tipo di carattere: Non Corsivo

ha formattato: Tipo di carattere: Non Corsivo ha formattato: Tipo di carattere: Non Corsivo ha formattato: Tipo di carattere: Non Corsivo

ha formattato: Tipo di carattere: Non Corsivo ha formattato: Tipo di carattere: Non Corsivo

[revised manuscript text omitted]
 geo-statistic approach herein proposed provides a tool to analyze analyze whether, and how far, people
- 845 behaviorsbehaviours are different from those of common days. Actually, a sample far larger than two years would be beneficial to make such statistics more reliable, since alarms interestinvolve a few days in a year. It is worth noting that in the analyzedanalysed area the risk perception towards the secondary network is almost absent, as well as a capillary local warning system. Flood risk perception is mainly related to the primary hydrographic network (i.e. Mella River). Therefore, in the regardings of the specific test case, in this research the possibility of drastic changes in human behaviorbehaviour during heavy rainfall alarms are not expected. Indeed, people virtuous behaviors are usually the result of extensive campaigns to
- 850 rainfall alarms are not expected. Indeed, people virtuous behaviors behaviours are usually the result of extensive campaigns to raise public awareness against flood risk, coupled with trusted and effective warning systems. Hence, a final objective of the ongoing research will take into consideration the effectiveness of the non-structure practices that will be adopted to mitigate flood risk in the test watershed.

**6 Acknowledgements**

[revised manuscript text omitted]

ha formattato: Inglese (Regno Unito)

ha formattato: Tipo di carattere: (Predefinito) Times New Roman